# Stream: A Generalized Continual Learning Benchmark and Baseline

## Abstract

In a typical Continual Learning (CL) setting, the goal is to learn a sequence of tasks that are presented once while maintaining performance on all previously learned tasks. Current state-of-the-art approaches require the task identity during training to mitigate forgetting, whereas General Continual Learning (GCL) evaluates the ability to learn the sequence of tasks without their identity. We find that GCL methods (Buzzega et al., 2020a; Wortsman et al., 2020) ignore the *domain gap* between two consecutive tasks ('learning-gap') and, as a result, often fail under more challenging scenarios. Motivated by a *learner* that needs to generalize across modalities and tasks, we propose a challenging GCL benchmark: the multi-modal **Stream**. Our benchmark provides a method to construct a sequence of tasks with varying *learning-gaps* from Vision and Text datasets. We perform a systematic analysis of meta-training statistics from the literature that are used to identify novel tasks, to find that they correlate to the learning-gap. Inspired by biological mechanisms of learning in mammals, we propose a baseline method to achieve GCL on *Stream*: $\alpha$**MetaSup**, which uses a 'dummy' Stream to train a Transformer model to identify novel task transitions ('surprise'). The trained Transformer is then used as an auxiliary novelty detector to a learner in the benchmark Stream. We show how $\alpha$MetaSup can augment existing CL methods that use rehearsal memory and improve their performance by as much as 10.5% AUC thereby outperforming 7 SOTA GCL baselines.

## 1 Introduction

Continual Learning (CL) approaches reduce *catastrophic forgetting* that neural networks suffer when learning a sequential, non-i.i.d. stream of tasks. CL, typically, defines a discrete *task boundary* between two adjacent tasks in the data stream where the difficulty of the stream of tasks can be quantified by the learning-gap between the two adjacent tasks.

Current CL benchmarks create a sequence of tasks by applying transformations (*e.g.* Permuted/Rotated MNIST), or splitting a standardized dataset (*e.g.* Split-CIFAR), in which the learning-gaps are close to uniform. However, in a more general and challenging setting there exists a mixture of large (domain shift) and small (distribution shift) gaps. For example, there can be a difference in the learning difficulty based on the order in which a sequence of three tasks is presented. For example, consider the task of classifying cat breeds, followed by English dog breeds (big gap from cats), and finally American dog breeds (smaller gap from English dogs). The learning-gap between consecutive tasks will influence the difficulty in mitigating forgetting (Lange et al., 2019), such as the difference between learning dog breeds and cross-species breeds. We refer to them as the **learning-gaps** in this work. Therefore, previous evaluation settings where the learning-gap is uniform may not be adequate to evaluate the generalization of a proposed method in a realistic scenario (Davari et al., 2022).

GCL approaches (Riemer et al., 2018; Zeno et al., 2018a) do not require the knowledge of task-boundaries (Riemer et al., 2018; Zeno et al., 2018a), but fail to perform in challenging settings where there is a long stream of tasks (Fostiropoulos et al., 2023). Current state-of-the-art (Wortsman et al., 2020; Kirichenko et al., 2021; Aljundi et al., 2019a) implements extra machinery to identify task transitions and then use a standard CL approach such as (Kirkpatrick et al., 2017; Rebuffi et al., 2016; Chaudhry et al., 2018) to mitigate forgetting. When task transitions are incorrectly detected, such methods fail catastrophically, where a CL method is applied incorrectly.

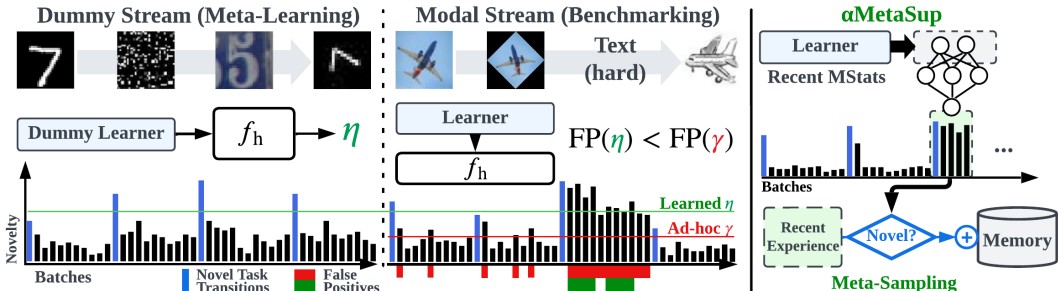

Figure 1: Illustration of our Stream benchmark that is used as a meta-learning objective in identifying novelty. **Left:** our meta-learning process uses a 'dummy' learner that does not attempt to mitigate forgetting. It is exposed to a stream of heterogeneous tasks (with different 'learning-gaps'), just like the actual learner. By monitoring transitions to novel tasks (blue bars) over time, the dummy learner, in its simplest form, adjusts a dynamic novelty threshold $\eta$ (**green**) to maximize future novelty prediction. **Middle:** compared to the previously used static ad-hoc threshold $\gamma$ (**red**), our meta-learning process improves the false positive rate (*i.e.* **area** under the novelty bars). Yet, using a threshold can still fail for difficult tasks (*e.g.* 'Text') where difficult samples are hard to distinguish from novelty. **Right**: as a baseline method to Stream, we use the same meta-learning process to train a Transformer model on a window of meta-learning statistics that is more robust to heterogeneous learning-gaps than the linear threshold $\eta$. In addition to detecting transitions to novel tasks, our Meta-Sampling also updates the memory used by the main learner to mitigate forgetting.

Identifying a novel task is composed of two problems, first quantifying (*e.g.* via a statistical metric) and then classifying (*e.g.* by a classification threshold) novelty. Current methods of identifying novelty use meta-learning statistics such as loss, gradients, or the feature space. There is a lack of systematic evaluation of meta-learning metrics in how informative they are in predicting novelty, where their choice in many methods (Mundt et al., 2020; Aljundi et al., 2019a) is often ad-hoc. Additionally, the choice of a meta-learning statistic is orthogonal to the way it is applied, *e.g.* one could use the loss as opposed to the gradients for computing the exponential moving average (Zhu et al., 2022). Lastly, the threshold used to classify novelty is determined ad-hoc (Wortsman et al., 2020; Aljundi et al., 2019a) or empirically (Kirichenko et al., 2021), while few methods (Rios et al., 2022) provide a method to automatically determine such threshold. Additionally, the threshold is identified by evaluation on the same dataset, or split of, as the model was trained on. As such, when the same threshold is evaluated in novel settings, it fails to generalize.

Our work is motivated by the lack of principled GCL evaluation benchmarks for a multi-modal setting of text and vision. We first propose **Stream** as a way to construct infinitely long task sequences with non-uniform learning-gaps by a finite set of 'base datasets' used with synthetic transformations. The construction of Stream allows it to be used as an effective meta-learning objective in novelty detection that improves generalization. We systematically evaluate common metrics used to quantify novelty to find that many of them are uninformative when the difference between tasks is non-uniform, and current GCL methods in identifying novelty fail catastrophically. We propose a baseline novelty detection $\alpha$MetaSup to Stream, which is inspired by the biological mechanism of 'learning through play' in mammals (Wang, 2021), where we use a 'dummy' Stream to train a Transformer model in identifying novelty. Similar to the metacognitive mechanism of surprise (*i.e.* due to novelty) (Foster & Keane, 2019), we sample 'experiences' (data) prior to but not including the novel data batch (*Meta-Sampling*) as a baseline GCL approach to our benchmark.

Due to the page limitation, consequence of the format this work is published in, we focus on the key insights of our work in the main text while supporting experiments and analysis are attached in the supplementary. Our main contributions:

- We introduce *Stream*, a new GCL benchmark and a method to construct an *infinite* stream of tasks with non-uniform learning-gaps. We survey *Meta-Statistics* (Table 2) from the literature to show the novelty of our benchmark and when compared with previous benchmarks.

- We propose a baseline method to Stream $\alpha$*MetaSup*, and use Stream as a meta-training objective for a Transformer model that predicts novelty. We use $\alpha$MetaSup as a method to construct long-term memory that augments previous state-of-the-art CL methods.

- We use our method and 7 state-of-the-art baselines on a *multi-modal Stream* of 40 tasks. $\alpha$MetaSup augmented baselines are the only ones that perform better than random in our challenging scenarios where their performance improves by 9% on average.

## 2    RELATED WORKS

**General Continual Learning (GCL)** (Buzzega et al., 2020a; Wang et al., 2023) seeks to learn without requiring knowledge of task identity during both training and inference, a general case of CL where the model learns domain-incrementally (DIL) by remapping task classes to overlapping concepts (van de Ven et al., 2022). For reasons of brevity, we refer the reader to previous work (Lange et al., 2019; van de Ven & Tolias, 2019) on the formal definition of different CL scenarios (*e.g.* CIL/TIL). **GCL Benchmarks** evaluate a method by transforming an existing dataset into several tasks, either by splitting the dataset classes Split-CIFAR (Krizhevsky et al., 2009; Wortsman et al., 2020), applying rotations Rotated-MNIST (LeCun, 1998; Wortsman et al., 2020), or permuting the features Permuted-MNIST (LeCun, 1998; Buzzega et al., 2020a). Such dataset and more recent dataset (Lomonaco & Maltoni, 2017; Bornschein et al., 2022; Lin et al., 2021; Sagawa et al., 2022) construct task sequences with uniform learning-gaps between tasks. Most similar to our work, the iBlurry and CLiMB benchmark (Koh et al., 2022; Srinivasan et al., 2022) motivates the need for non-uniform learning-gaps that previous benchmarks fail to address. iBlurry splits classes of 4 datasets into 25 tasks where a class can belong to more than one task. It can be argued that their approach does not meet CL where i.i.d samples from a given class are repeated between tasks, whereas we do not repeat a sample for a given class without first applying a transformation to it for a given task. CLiMB constructs the data sequence from vision and language datasets with non-uniform learning-gaps, contrary to our work, they supported a limited number of 13 tasks. Similarly to the aforementioned benchmarks, Stream applies transformations to the original datasets to create new tasks. On the contrary, we perform the transformations in the feature space where there are more degrees of freedom to create 'infinitely' many tasks with varying difficulty. Additionally, our method is able to project multiple datasets of multiple modalities on a common feature space and evaluate a learner without the influence of the feature extraction method.

**Novelty quantification and detection** is the goal of several ML fields (Mundt et al., 2020) such as *Out-Of-Distribution* (OOD) detection (Yang et al., 2021). GCL novelty detection is performed in an online manner to detect *task transitions* to mitigate forgetting as opposed to in an offline manner between a train and the test set to identify outliers. Novelty detection methods and GCL approaches (Aljundi et al., 2019a; Ardywibowo et al., 2022; Wortsman et al., 2020; Zhu et al., 2022) *quantify* novelty by meta-training statistics to infer a task transition that is then combined with a classification method, *e.g.* variance from the exponential weighted moving average of loss (Zhu et al., 2022). We can evaluate a novelty detection method using the F1 score to account for the low positive rate of task transitions compared to the number of batches we evaluate. A high false-positive rate can be catastrophic (*i.e.* applying an additional penalty or oversampling of a given task).

**Novelty Identification** in an online manner is performed using heuristic methods (Aljundi et al., 2019a; Wortsman et al., 2020; Zhu et al., 2022) or adaptive methods such as **Temporal Anomaly detection** (Sipple, 2020; Lindemann et al., 2021) using LSTMs. Bayesian approaches can be used to detect changes based on distribution shifts using a heuristic threshold (Kirichenko et al., 2021; Lee et al., 2020; Ardywibowo et al., 2022) or agnostic to a threshold (Zeno et al., 2018a;b; 2021; Li et al., 2021; Kessler et al., 2021; Itti & Baldi, 2009; Rios et al., 2022). The CL method is then applied when novelty is detected, where the key contribution of the method is in novelty identification. We find that Bayesian approaches fail to generalize to larger sequences of tasks, as they maintain statistics over a growing number of data that becomes intractable. Additionally, when the threshold for novelty is chosen ad-hoc, it fails to generalize. Motivated by the inadaptability of previous methods to perform to our challenging benchmark, we develop a meta-learning approach where we train an adaptive classifier (*i.e.* Transformer) to infer novelty in a dummy Stream. The choice of the architecture of the novelty identification model is orthogonal to the main contribution of our work. We evaluate several models, including the ones mentioned, to find that our Transformer model has improved generalization.

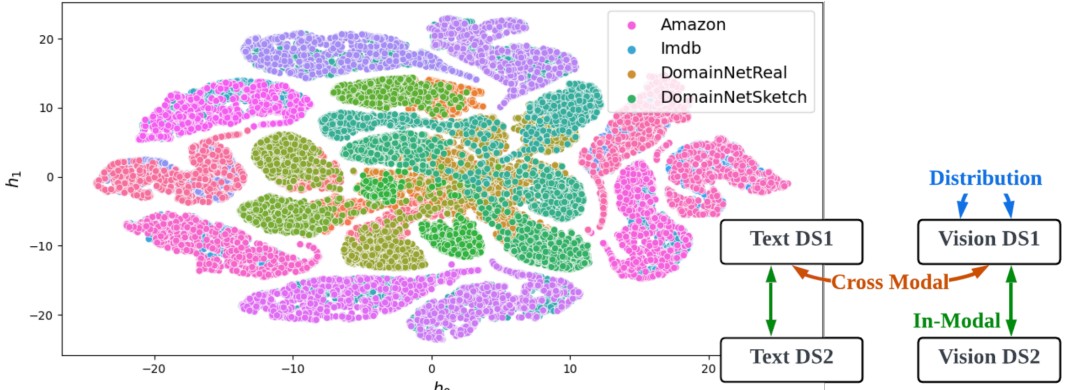

Figure 2: **Left**, we perform T-SNE (Van der Maaten & Hinton, 2008) on the feature vectors from the tasks in S-Modal to visualize commonalities between them. We assign a color to each vector from a continuous spectrum based on the alphabetical order of the dataset and transformation type. It can be observed that the color for a given dataset changes on a gradient (*e.g.* 'Amazon' on the left outer side) and for a given dataset different modalities produce distinct clusters. Additionally, for similar modalities, there is a high overlap, *e.g.*, 'Amazon' (pink) overshadows 'IMDb' (blue). The Vision tasks (Real, Sketch) are clustered in the center, while Text datasets (Amazon, IMDb) are on the outer side. Our visualization confirms that the Stream learning-gaps are similar but distinct enough that T-SNE can identify their commonalities. **Right**, our transformations (Appendix A.1) provide three types of learning-gaps: *Cross* and *In-Modal Gaps* are transitions between tasks that have different base datasets. *Distribution Gap* is the transition between tasks on the same base dataset and modality.

## 3 STREAM

**Stream** is a sequence of tasks generated by projecting multi-modal Vision and Text datasets on the same feature space and augmenting them **within** the feature space to synthetically generate an arbitrarily long sequence of tasks. By augmenting the tasks in the feature space, our method can control the difference of the learning-gaps as it provides more degrees of freedom, a visualization of the learning-gaps as well as their categories can be found in Figure 2. Projecting all datasets on the same feature space has the advantage that we can use a single learner and evaluate how well it performs in a challenging setting while also removing the bias from the feature extraction process, *i.e.* we are ablating the performance of the learning algorithm and not the robustness of the feature extractor, while at the same time it is computationally tractable, for example we can evaluate a large sequence of tasks in the reduced dimension space.

### 3.1 TASK-CONSTRUCTION

Stream uses transformations $T$ to create a large sequence of 'novel' tasks with respect to the learner but using a limited set of datasets $D_{\text{base}}$. Transformations provide a method to control the learning-gap between tasks while not altering the difficulty of the dataset. For example, a learner should be able to perform equivalently when trained in isolation on both the transformed and un-transformed datasets, but will perform differently when trained on the same set of tasks in a different order.

**Task Sequence** can be defined as a list of data transformations where $T_i \in \{T_1, \ldots, T_n\}$ is applied on a list of datasets $D_{\text{base}}$. $T_i$ is uniquely defined by the random seed. **Stream** is able to produce infinitely long task sequences from a limited set of $D_{\text{base}}$ by varying the random seeds. We construct and evaluate three Streams: S-Num, S-Vis, and S-Modal; Table 1. Due to the page limit, we show the details of the Stream Transformations in Appendix A.1.

### 3.2 CONCEPT MAPPING

For Stream, there can exist base datasets that have one-to-one correspondences between their classes such as the numbers of MNIST and SVHN (S-Num). While for more complex settings, *e.g.* Text → Vision, the same is not true. It is an open problem on how to design a learner without a priori

Table 1: A 'Stream' can be constructed from datasets $D_{\text{base}}$ by applying Stream Transformations. We compare our dataset with standard CL benchmarks PMNIST and Split-CIFAR100 (where none or only a single transformation is applied). We denote the difficulty of each base dataset $\overline{\text{AUC}}_{D_{\text{base}}}$ for a learner trained on each task independently (no CL). We report the theoretical Upper-Bound performance $\overline{\text{AUC}}_{\text{UB}}$ that a learner can achieve on the given Stream in GCL as the mean of $\overline{\text{AUC}}_{D_{\text{base}}}$. We find that S-Modal is the most challenging Stream. The low variance ($\pm\sigma_{\text{AUC}}$) suggests that the Stream transformations do not affect the task difficulty.

| Stream | $D_{\text{base}}$ | $N$ Samples | $\overline{\text{AUC}}_{D_{\text{base}}}$ | $\overline{\text{AUC}}_{\text{UB}}$ |
|---|---|---|---|---|
| SplitCIFAR | CIFAR$_{1,\ldots,10}$ | 6,000 each | - | 0.969 |
| PMNIST | MNIST (LeCun, 1998) | 60,000 | - | 0.998 |
| S-Num | MNIST (LeCun, 1998) | 60,000 | $0.998 \pm 0.001$ | 0.959 |
| | SVHN (Netzer et al., 2011) | 99,289 | $0.921 \pm 0.010$ | |
| S-Vis | CIFAR-10 (Krizhevsky et al., 2009) | 60,000 | $0.969 \pm 0.004$ | 0.950 |
| | CINIC-10 (Darlow et al., 2018) | 270,000 | $0.930 \pm 0.006$ | |
| S-Modal | Real (Peng et al., 2019) | 569,000 | $0.980 \pm 0.001$ | **0.813** |
| | Sketch (Peng et al., 2019) | 569,000 | $0.906 \pm 0.002$ | |
| | IMDb (Maas et al., 2011) | 50,000 | $0.687 \pm 0.050$ | |
| | Amazon (Ni et al., 2019) | >20,000,000 | $\mathbf{0.677 \pm 0.076}$ | |

knowledge of the tasks and their classes. Domain-Incremental Learning (DIL) projects multiple classes to 'concepts', where for example a dog and a cat could be mapped to the same concept 'animal'. We observe that the semantic similarity of classes mapped to the same concept does not have a negative impact on the performance (see supplementary Sec. E.1), such as mapping an onion and a snail to the same concept as opposed to an onion and a carrot. CL benchmark that evaluates the ability to learn concepts rather than objects is supported by previous work (Lin et al., 2022).

**S-Modal Concept Mapping** The original DomainNet-Real ('Real') and DomainNet-Sketch ('Sketch') datasets (Peng et al., 2019) contain 345 classes. We use K-Means clustering on the CLIP (Radford et al., 2021b) features extracted from the class name textual description. Thus, we group the 345 DomainNet classes into 10 concepts for Real and Sketch datasets. The numbers of train classes in S-Modal for Real, Sketch, IMDb, and Amazon tasks are 10, 10, 10 and 5, respectively. We provide additional information on the mapping in the supplementary material. As the label distribution between $D_{\text{base}}$ can be imbalanced (*e.g.* Real and Sketch 10 classes and Amazon 5 classes), we evaluate the performance of a learner using the AUC score.

## 4 LEARNING-GAP

Several previous works have investigated the stability-gap (Lange et al., 2023) and task-order (Bell & Lawrence, 2022; Lange et al., 2019) or as we commonly refer to in this work as the *learning-gap* between tasks. Previous works in GCL have proposed extracting model training statistics, such as loss and gradient, for evaluating the learning-gap and identifying novel tasks. We categorize the Stream learning-gaps in Figure 2 (right) and evaluate them in Section 5.1. We use Stream as a novel setting to evaluate previous GCL works under non-uniform learning-gaps.

### 4.1 PRELIMINARIES

**Meta-Statistics** (MStats) are meta-training statistical metrics, such as mean loss value, that can be extracted from a neural network. We survey the most widely used statistical metrics from literature, and provide six categories in Table 2: Loss (Aljundi et al., 2019a; Zhu et al., 2022), Gradient (Huang et al., 2021), Fisher Information (Park et al., 2022), Feature (Tack et al., 2020), Uncertainty (Wortsman et al., 2020), and Energy (Liu et al., 2020) that are used for both OOD detection and CL. These metrics can result in lists of tensors with different dimensions (*e.g.* list of gradients from neural network layers), and they are not directly applicable to novelty detection methods. Previous works, perform *dimensionality reduction* (see supplementary Sec. C.1) to aggregate them as an intermediate

step before using them with their method. We evaluate all 6 MStats with 63 evaluation protocols on their discriminative power to identify novel tasks, in Section 5.2. Our work is the first to perform a systematic evaluation and observe that novelty detection is sensitive to both the MStats used and their reduction methods.

### 4.2 ADAPTIVE META-LEARNING

Previous works (Aljundi et al., 2019a; Ardywibowo et al., 2022; Wortsman et al., 2020; Zhu et al., 2022) use an ad-hoc fixed hyperparameter as a threshold to classify whether MStats indicate a novel task that fails to generalize, as we find in this work. We propose the use of a **dummy Stream** constructed agnostic to the main task sequence used during the training of the learner to the benchmark. For example, we find a threshold ($\eta$) through meta-learning on S-Num (from Table 1) by a heuristic meta-learner $f_h$ and use the same $\eta$ to evaluate the learner on S-Modal; Figure 1 (middle). We use the same neural network architecture (*i.e.* a dummy learner) during meta-learning as the benchmark training, but without a mechanism to mitigate forgetting and observe the learner's behavior when exposed to novel tasks. Different methods

Table 2: Summary of Meta-Statistics commonly used in CL and OOD literature for novelty detection. Where $f_\theta$ is the model parameterized by $\theta$, $\mathcal{L}$ the loss function, $\hat{y}$ a pseudo-label, $\phi$ the set of feature vectors, $\mathcal{H}$ the entropy function. It can be observed that metrics share similarities where, for example, Fisher Info is a special case of Gradient. We find that simpler statistics often outperform their complex counterparts in GCL.

| MStats ($\tau$) | Definition |
|---|---|
| Loss ($\tau_L$) | $\mathcal{L}(f_\theta(x), y)$ |
| Gradient ($\tau_G$) | $\nabla_\theta \mathcal{L}(f_\theta(x), y)$ |
| Fisher Info. ($\tau_{FI}$) | $\theta + [\nabla_\theta \mathcal{L}(f_\theta(x), \hat{y})]^2$ |
| Feature ($\tau_F$) | $\phi(x)$ |
| Uncertainty ($\tau_U$) | $\mathcal{H}(f_\theta(x))$ |
| Energy ($\tau_E$) | $-\log \sum \exp(f_\theta(x))$ |

can be used to predict novelty, such as the Running Mean (Aljundi et al., 2019a), Bayesian Inference (Kirichenko et al., 2021) or Contrastive (Aljundi et al., 2019b). Such methods are used with a fixed threshold value $\eta$ to detect novelty. For example, a method can use the variance from the running statistics and trigger novelty when it exceeds a predefined threshold. We evaluate all methods (see supplementary Sec. C.1) to find that Running Mean works the best among the heuristic-based methods and denote it as $f_h$ in the main text. However, we find that a heuristic method fails in more challenging multi-modal settings, *e.g.* S-Modal. As such, we introduce a baseline method to help further research in the field.

$\alpha$**MetaSup** can be used to augment any CL method that utilizes a memory of a sort. $\alpha$MetaSup replaces the inflexible $f_h$ from previous work with an adaptive neural network $f_\alpha$ trained on time window $w$ of MStats $\tau_{i-w:i}$ such that:

$$\max_{f_\alpha} \mathbb{E}[P(f_\alpha(\tau_{i-w:i})| \, i \rightarrow \texttt{Surprise})] \tag{1}$$

$\alpha$MetaSup computes the novelty probability based on a window of MStats as opposed to only the most recent batch. Additionally, the highly parameterized model $f_\alpha$ can learn a non-linear relationship between novelty and MStats that 'adapts' to the non-uniform learning gaps and as shown in Figure 1 (right). The choice of the backbone $f_\alpha$ is flexible. We evaluate a Transformer (Vaswani et al., 2017) ($f_{gpt}$) and an LSTM ($f_{lstm}$) to find that $f_{gpt}$ generalizes the best among backbones, datasets, and MStats. We present the performance of $f_{gpt}$ in Table 4 and the ablation of the $f_\alpha$ architecture in the supplementary.

We use $\tau$ as a metric to quantify novelty and $f_\alpha$ to identify novelty based on a context window. The main contribution of $\alpha$MetaSup is on the learning process, where, similar to the flexible choice of $\tau$ and $f_\alpha$, the method to mitigate forgetting is a design choice. A CL method can be used when a novel task is detected to mitigate forgetting. We find that current CL methods are not robust to false positives where a CL method is incorrectly applied within the same task. As such, we provide an improvement to CL methods that utilize memory where Meta-Sampling stores long-term memory samples prior to 'surprise' at step $i$ but not including data at $i$. Meta-Sampling is robust to false positives as the data that triggered a false-positive response are not stored in long-term memory. We find similarities to the biological mechanism of learning, where experiences leading to a traumatic event are vivid, while the traumatic event is nonmemorable (Foster & Keane, 2019). We provide the details of Meta-Sampling and compare our method with additional related works in the supplementary Sec. D and B respectively.

Table 3: Analysis on the informativeness of Loss MStats ($\tau_L$) in predicting novelty. We report the raw numerical value of MStats, and the ones computed by heuristic meta-learner $f_h$. We report the value of a 'Novelty' metric from 1 step prior to a novel task $\rightarrow$ when a novel task appears. The informativeness can be evaluated by the numerical difference between transition values where $f_h$ provides a larger difference (*e.g.* $-0.02 \rightarrow 4.27$ compared to $0.16 \rightarrow 0.16$). We find that the informativeness corresponds to the four categories of learning gaps from Figure 2.

| Novelty | Cross-Modal | In-Modal | Distribution |
|---------|-------------|----------|--------------|
| $f_h(\tau_L)$ | $-0.06 \rightarrow 34.86$ | $-0.18 \rightarrow 15.38$ | $-0.02 \rightarrow 19.17$ |
| $\tau_L$ | $1.08 \rightarrow 5.25$ | $1.05 \rightarrow 3.14$ | $1.04 \rightarrow 3.37$ |

## 5 EXPERIMENTS

In Section 5.1, we evaluate the Stream learning-gap by the type and size of the task transition. In Section 5.2, we evaluate the Meta-Statistics and their usefulness in novelty detection and show how a heuristic function $f_h$ from Section 4.2 can generalize poorly. Last, in Section 5.3, we show the improvement our meta-learning process using Stream can provide to existing novelty detection methods, while the $\alpha$MetaSup augmented method is the only one that can perform in our most challenging setting, a multi-modal Stream. We provide additional experiments on time efficiency, ablation studies, and details of the experimental setup in our supplementary Sec. E and F. We present and discuss the main results in the remainder of this section.

### 5.1 STREAM LEARNING-GAPS

We use Forward Transfer (FWT) (Lopez-Paz & Ranzato, 2017) in the AUC score to evaluate the similarity between two consecutive tasks ($t_a \rightarrow t_b$) and as a proxy to quantify the learning-gap. FWT is the performance of a learner on an unobserved task $t_b$ after training on some task $t_a$. Trivially, a larger FWT between two tasks signifies a higher similarity between them and a smaller learning-gap. For example, a learner can be trained on MNIST and evaluated on SVHN. We would expect the learner to perform on SVHN without being trained on the dataset as it is the colored version of MNIST. The same can not be said for the 'text' version of MNIST *i.e.* a Cross-Modal gap. We observe an FWT of 0.51 for Cross-Modal gaps, 0.76 for In-Modal Gaps, and 0.57 for Distribution Gaps (minimum transformation strengths). Our experiments have us conclude that learning in the feature space bears similarities to learning in the input space as the FWT agrees with our intuition. The advantage of Stream being applied in the feature space is that multiple modalities can be projected on the same feature space. A full table comparing these learning-gaps at different levels of transformation strengths can be found in Appendix E.1.

### 5.2 META-LEARNING

We first evaluate how a heuristic novelty function $f_h$ can improve the discriminability of novel tasks from all learning-gaps in Table 3. Compared to using MStats as novelty scores directly, $f_h$ drastically increases the numerical difference between a learned task and a novel task. The magnitude difference between learning-gaps suggests that the Cross-Modal Gap is more distinguishable than In-Modal and Distribution Gaps, and a fixed threshold may be hard to generalize to all learning-gaps. A full table can be found in Appendix Table 6.

Next, we evaluate the generalization of $f_h$ using a fixed threshold acquired from the meta-learner. We survey and several $\tau$ and present a representative sample in Table 4. We collect $\tau$ from 20 randomly initialized task sequences and for 5 different datasets. We evaluate the performance of a metric $\tau$ in its discriminative power to predict novelty. We find that the performance of a given $\tau$ generalizes among datasets, while it is influenced by the dimensionality reduction method. Next, we evaluate the ability of the novelty threshold ($\eta$) to generalize between S-* ($\rightarrow$) datasets, where the lower variance indicates higher similarity on the threshold value between the two datasets. The results have us conclude that $\tau_G$ is informative, while it is expected to generalize better, as evaluated by the smaller difference in VMR (Dispersion Ratio (Cox & Lewis, 1966)). Additionally, we conclude that

Table 4: We evaluate the generalization of Meta-Statistics $\tau$ when used with a threshold heuristic ($f_h$ Section 4.2), a novelty detection method that is similar to (Wortsman et al., 2020; Zhu et al., 2022; Aljundi et al., 2019a). On the **left** we report the dispersion ratio ('VMR') of the learned threshold ($\eta$) at each dataset for 100 random task sequences. Additionally, we report the dispersion ratio for the two datasets combined ($\rightarrow$). A larger in-dataset VMR signifies a difficult task sequence for which a different threshold is identified at each iteration where Stream task sequence ('S-Vis') is an improvement to traditional benchmark Split-CIFAR100 ('S.C.'). On the **right** we evaluate the learned $\eta$ in predicting novelty on a held-out task sequence of 'S.C.' and apply the same threshold on 'S-Vis' to evaluate Novelty Detection ('F1'). When comparing between $\tau$ a smaller VMR and higher F1 signifies improved generalization. Our results show that lower dimensional $\tau$ are more noisy (higher $\rightarrow$ VMR for $\tau_L$ compared to $\tau_G$) while the *dimensionality reduction* can affect the generalization where $||\tau_G||_2$ outperforms $\overline{\tau_G}$. A heuristic method can generalize poorly for more complex $\tau$ that motivates us to develop an adaptive method, $\alpha$MetaSup. A full table of 63 evaluation protocols is provided in Supplementary Tables 10 to 13.

| $f_h(\tau)$ | Threshold - Dispersion Ratio (VMR) | | | Novelty Detection (F1) |
| | S.C. $\uparrow$ | S-Vis $\uparrow$ | S.C. $\rightarrow$ S-Vis $\downarrow$ | S.C. $\rightarrow$ S-Vis $\uparrow$ |
|---|---|---|---|---|
| $\tau_L$ | 0.187 | 0.721 | 1.444 | $0.947 \rightarrow 0.947$ |
| $\overline{\tau_G}^{\ddagger}$ | 0.192 | 0.583 | 0.866 | $0.947 \rightarrow 0.774$ |
| $||\tau_G||_2^{\dagger}$ | 0.144 | 0.497 | 0.801 | $0.842 \rightarrow 0.800$ |
| $\overline{\tau_G}^{\dagger}$ | 0.304 | 0.767 | 1.124 | $0.941 \rightarrow 0.750$ |
| $\overline{\tau_{FI}}^{\dagger}$ | 0.248 | 0.870 | 1.343 | $0.947 \rightarrow 0.593$ |
| $\overline{\tau_E}$ | 0.260 | 0.477 | 0.915 | $0.340 \rightarrow 0.533$ |
| $\overline{\tau_U}^{\dagger}$ | 0.310 | 0.505 | 1.111 | $0.271 \rightarrow 0.245$ |
| $||\tau_F||_2^{\ddagger}$ | 0.288 | 0.341 | 2.147 | $0.120 \rightarrow 0.558$ |

$\tau$ which are higher dimensional representations of other $\tau$ perform poorly than their counterparts *e.g.* $\tau_{FI}$ and when compared to $\tau_G$. We hypothesize that aggregate effects over the high-dimensional space can lead to noisy signals for novelty, while scalars such as $\tau_L, \tau_E, \tau_U$ generalize poorly or have high VMR, making them challenging to use.

We use $\tau_G$ (low VMR) to evaluate the generalization of an adaptive method $f_\alpha$ to identify novelty and when compared with a heuristic $f_h$. We train a dummy learner on S-Num and evaluate on a ResNet and Residual MLP on S-Modal and S-Vis (cross-dataset evaluation) in a GCL setting. We find that $f_h$ generalizes poorly in the new setting where it now performs with 0.738 average F1 Score (Table 9) compared to 0.872 from the in-dataset evaluation setting (train and evaluate the dummy learner on the same Stream). Contrarily, $f_{gpt}$ performs with 0.861 F1 Score for the same setting (Table 9), a 16.7% improvement in performance compared to $f_h$. We conclude that an adaptive novelty detection method generalizes better across datasets. Additional experiments on $f_h$, $\alpha$MetaSup and full tables of MStats evaluation are in supplementary E.3 and Tables 10 to 13.

## 5.3 STREAM BENCHMARK GCL

We present our results of a multi-modal Stream (S-Modal) as a challenging benchmark to evaluate the generalization of a learner in Table 5. We evaluate current state-of-the-arts that do not require task identity to mitigate forgetting, and find that they can perform for easier Stream but fail catastrophically when evaluated on a set of tasks with large learning-gaps, such as S-Modal. To provide a baseline result for future studies on Stream benchmarks, we augment ER (Riemer et al., 2018) with a state-of-the-art novelty detection method and Meta-Sampling ('MSamp', Ours). We find that augmenting ER with a novelty identification method (*e.g.* $f_{OneShot}$) and Meta-Sampling improves performance by as much as 10.5% and 9.6% for easier dataset S-Vis and S-Num. We conclude that Meta-Sampling and Novelty identification are important in mitigating forgetting. However, the fixed threshold novelty detection methods (*i.e.* middle portion of Table 5) perform poorly for 'S-Modal'. As such, we replace the novelty identification component of the previous methods where we use $\alpha$MetaSup trained on a dummy Stream composed of different datasets that we then evaluate on Stream. For example, we train $\alpha$MetaSup with a dummy learner on S-Num and evaluate when used with a new learner on S-Modal. We find that the $\alpha$MetaSup augmented baseline ER performs 8% better. $\alpha$MetaSup is flexible and

can be used with any CL method that has a memory component. We conclude that adaptive methods will be required for more challenging settings where heuristics fail. Lastly, for a GCL setting, a meta-learning method will be required where $\alpha$MetaSup has flexible design components that can be used to augment existing approaches.

## 6 DISCUSSION

There is currently no work that introduces a method to control and construct a stream of data with variable learning-gaps in real-world applications. Additionally, current Meta-Statistics can either be impractical to obtain in an online fashion (Jacobian), too noisy in predicting a surprise (Fisher Information), or uninformative (Features), and therefore they can be difficult to use in predicting a learning-gap for GCL. Previous work, and our work, still relies on artificial transformations to generate learning-gaps of variable size. We call for future work on improved statistics to improve novelty detection's robustness in GCL.

Little work exists on the evaluation of false-positive responses to novelty. We motivate the development of evaluation protocols to understand the catastrophic forgetting of GCL methods when incorrectly identifying novelty.

## 7 CONCLUSION

In this work, we propose Stream as a method to construct infinitely long task sequences with varying learning-gaps, which is an evaluation scenario commonly overlooked by previous GCL works. We perform a systematic review of Meta-Statistics that are used to identify novel tasks in an online manner to evaluate the diversity of the task sequences constructed by Stream. We proposed $\alpha$MetaSup a biologically inspired meta-learning novelty detection method that is robust to false positives compared to previous work. $\alpha$MetaSup is flexible where the specific meta-training statistic and the CL method used to mitigate forgetting are design choices. We exhaustively evaluate several design choices and the components of our learning method in ablation studies to conclude that $\alpha$MetaSup is the only method that can perform well in a challenging multi-modal setting constructed with Stream. Our benchmark reveals weaknesses of existing GCL methods where we find that meta-learning would be required to mitigate forgetting. The flexible construction of Stream can provide a method to construct such a meta-learning objective and evaluate a GCL method.

Table 5: **Top** rows; we report the AUC performance of SOTA rehearsal baselines ER (Riemer et al., 2018), DER++ (Buzzega et al., 2020a), CLS-ER (Arani et al., 2022) on Stream datasets. All baselines perform similarly to each other and perform close to random (0.5) for multi-modal Stream ('S-Modal'). **Middle** rows; we improve ER by 10% when we augment with Meta-Sampling (Ours), when computed as the average of the novelty detection methods $f_E(\tau_E)$ (Liu et al., 2020), $f_{\text{Peak}}(\tau_L)$ (Aljundi et al., 2019a), $f_{\text{OneShot}}(\tau_U)$ (Wortsman et al., 2020), $f_{\text{EWMA}}(\tau_L)$ (Zhu et al., 2022) with a fixed threshold. **Last** row; are the mean performance of our adaptive novelty detection method $\alpha$MetaSup. We conclude that all components of $\alpha$MetaSup improve the existing CL baseline (ER) while $\alpha$MetaSup is the only method that performs at S-Modal.

| Method | S-Vis | S-Num | S-Modal |
|---|---|---|---|
| DER++ | 0.735 | 0.894 | 0.565 |
| CLS-ER | 0.741 | 0.891 | 0.555 |
| ER | 0.724 | 0.908 | 0.555 |
| **Base Sampling** | 0.734 | 0.898 | 0.558 |
| $f_E(\tau_E)$ | 0.778 | 0.908 | 0.564 |
| $f_{\text{Peak}}(\tau_L)$ | 0.806 | 0.901 | 0.562 |
| $f_{\text{OneShot}}(\tau_U)$ | 0.770 | 0.915 | 0.564 |
| $f_{\text{EWMA}}(\tau_L)$ | 0.783 | 0.901 | 0.562 |
| $\overline{f_h}$ + **Meta-Samp.** | **0.784** | 0.907 | 0.563 |
| $\alpha$MetaSup + ER | 0.763 | **0.923** | **0.603** |

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

# A  STREAM DATASET

We use publicly available datasets previously used for Continual Learn (CL) and construct three Stream task sequences, S-Num, S-Vis, and S-Modal that we use for our experiments in the main text. For task transformations, we perform 2-D rotation transformation and channel-wise permutation on the features extracted by the first convolutional layer in a pre-trained and frozen ResNet-18, as discussed in Sec 3.1 in the main text.

**S-Num** is the Stream task sequence generated from MNIST (LeCun, 1998) and SVHN (Netzer et al., 2011) as 'base datasets' ($D_{\text{base}}$). S-Num is an improvement to the commonly used PMNIST benchmark, where a diverse sequence is an improved meta-learning objective to a 'dummy learner' (Table 3 in the main text). We use S-Num for training $\alpha$MetaSup which we then use for our more challenging vision (S-Vis) and multi-modal Stream experiments (S-Modal). When S-Num is used as a meta-training objective, on the dummy learner (Residual MLP (Touvron et al., 2021)) of S-Modal, we resize SVHN to 28x28, transform to grayscale, and flatten to 784-D vector. For the dummy learner (ResNet-18 (He et al., 2015)) of S-Vis, we resize MNIST to 32x32 and convert it to color. For all the experiments for which we report results for S-Num, we use the first variant of S-Num (Residual MLP) unless specified otherwise.

**S-Vis** is the Stream benchmark generated from CIFAR-10 (Krizhevsky et al., 2009) and CINIC-10 (Darlow et al., 2018) as 'base datasets'. CINIC-10 is a subset of ImageNet for the classes present in CIFAR-10. We use CINIC-10 to create tasks where there are domain shifts from CIFAR-10. S-Vis is a challenging alternative to SplitCIFAR.

**S-Modal** is a multi-modal GCL benchmark we include in Stream. The task sequence is generated from two Vision (Real, Sketch from DomainNet (Peng et al., 2019)) and two Text (IMDB (Maas et al., 2011), Amazon (Ni et al., 2019)) datasets. Contrary to S-Num and S-Vis, S-Modal does not have a 1-to-1 correspondence between the classes of the datasets, and as such we train a learner with concept mapping. We map the 345 DomainNet classes into 10 concepts, while the classes for the remaining datasets are unaffected (Table 13 and Table 14). We use a pre-trained ViT (Dosovitskiy et al., 2020) and GPT-2 (Radford et al., 2019) to embed Vision and Text datasets into a common 768-D feature space. The goal of the methodology is to ablate the efficiency of the GCL algorithm applied to the learner without the computational burden or bias of the feature extractor. Permutation and rotation transformations are performed in the 768-D feature space. We use a Residual MLP (Touvron et al., 2021) as a learner.

## A.1  STREAM TRANSFORMATIONS AND LEARNING-GAPS

**Stream Transformations.** We propose two types of transformations to construct Stream task sequences. **Permutation Transformations** $T_{\text{perm}}$ re-order the feature space but do not apply to data for which there are interdependencies among features. For example, for images and text, when permuting pixels or words, spatial dependencies will be perturbed and consequently increase dataset difficulty (Ivan, 2019). On the contrary, we permute across embedded dimensions of pre-trained features that allow us to generate novel tasks of the same difficulty. Moreover, permuting on the embedding space for modalities with few features such as images (3-channels) increases the number of possible transformations from 3! to $N$! (N-channel pre-trained features). **Rotation Transformations** $T_{\text{rot}}$ for high-dimensional feature spaces can have several interpretations (Aguilera & Pérez-Aguila, 2004). For Stream, we first project the feature Tensor to a 2-dimensional plane, then perform a 2-D rotation, and finally project the tensor back to the original space, thereby also preserving feature inter-dependencies. An illustration of feature-wise permutation and vector rotation is provided in Figure 3.

**Learning-Gap Categories.** As shown in previous work (Lange et al., 2019) the similarity between two sequential tasks can influence the forgetting a learner experiences. The goal of Stream is to provide a method to control for the similarity between 'learning-gaps'. Based on the transition between two adjacent tasks $t_a, t_b$ in Stream, we design our transformations to provide four types of 'learning-gaps', as provided in Figure 3 in the main text.

- **Distribution Gap** $t_{\text{MNIST}_1} \xleftrightarrow{rot} t_{\text{MNIST}_2}$; where $t_a, t_b$ are from the same dataset and permutation, with different degrees of rotation.

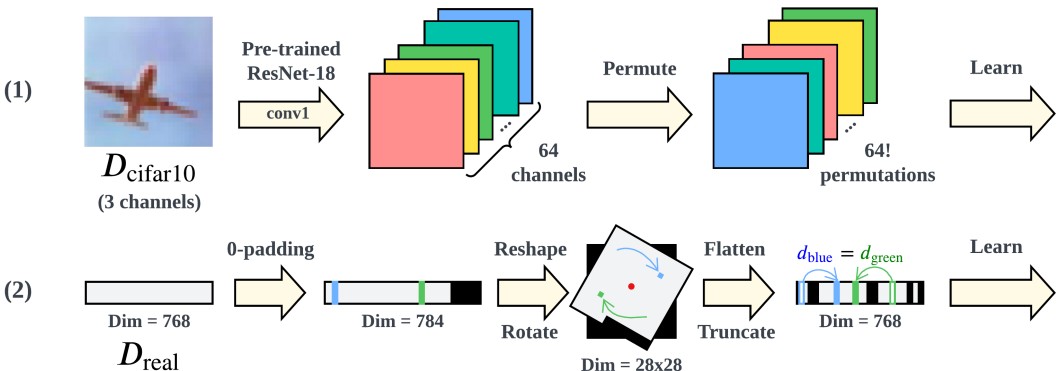

Figure 3: Illustration of the permutation $T_{\text{perm}}$ (1) and rotation $T_{\text{rot}}$ (2) transformations introduced by Stream. The application of the transformations in the feature space provides more degrees of freedom than transformations in the input space (*e.g.* PMNIST). **Top**, for a 3-channel image (1) from Cifar10 ($D_{\text{cifar10}}$) we use the pre-trained convolutional features of the first layer from a ResNet-18 to increase the number of unique transformations from $3! = 6$ to $64! \approx e^{89}$. A distinct task is created by a unique permutation seed and rotation angle. The transformations provided by Stream are applicable to multi-modal datasets where feature inter-dependencies such as in Text and Vision can exist. For example, vector rotation (**Bottom**) will perturb but not distort spatial inter-dependencies where there will be an equidistant shift in the vector space for features (*e.g.* $d_{\text{blue}} = d_{\text{green}}$).

- **In-Modal Gap** $t_{\text{MNIST}_1} \xleftrightarrow{rot} t_{\text{SVHN}_2}$; where $t_a, t_b$ are two distinct datasets of the same modality and permutation.

- **Cross-Modal Gap** $t_{\text{MNIST}} \longleftrightarrow t_{\text{IMDB}}$; where $t_a, t_b$ are of different modalities but for the same permutation.

- **Permutation Gap** $t_{\text{MNIST}_1} \xleftrightarrow{perm} t_{\text{MNIST}_2}$; where $t_a, t_b$ have different permutations. In the ablation study of learning-gaps (Table 7), we find that the Permutation Gap is similar to the Cross-Modal Gap in magnitude.

## A.2 DOCUMENTATION, LICENSE AND SOCIAL IMPACT

We will release the Stream Dataset repository as well as the documentation after the review process concludes. We include the documentation for creating Stream task sequences by a `TaskScheduler` so that users can construct any sequences with their custom base dataset. Our code and dataset of the extracted feature vectors are open-sourced and licensed under GPL-3.0. The datasets we used (MNIST, SVHN, CIFAR-10, CINIC-10, Amazon, IMDB, Real, and Sketch) to extract the feature vectors are not part of our work. For their usage and license, please refer to each respective dataset.

**Ethics and Social Impact.** All the datasets we use to construct the Stream are publicly available and are curated to not contain sensitive and private information. The users may choose to use their own dataset to synthesize Stream tasks. Stream offers the ML community an efficient way (*e.g.* using a small number of datasets) to construct a large sequence of tasks to evaluate the generalization of a learning algorithm in realistic scenarios.

## B ADDITIONAL RELATED WORKS

**Online Novelty Inference for GCL.** Previous GCL works infer when a novel task is introduced and apply a Continual Learning (CL) method. For example, a novel task can be inferred by the value of the loss function, where the loss has a smaller variance and a smaller value range between batches and before a novel task. A method can be used to determine the value of a meta-training statistic (*e.g.* loss) based on a threshold that detects novelty. Previous work (Aljundi et al., 2019a) uses the plateau and peak of the loss to detect novelty; $f_{\text{Peak}}(\tau_L)$. Similarly, (Zhu et al., 2022) maintains the exponential weighted moving average for the training loss, which is then used to identify novelty when the value exceeds a threshold; $f_{\text{EWMA}}(\tau_L)$. 'One-Shot' (Wortsman et al., 2020) uses a fixed

Table 6: Analysis of the linear separability of MStats Gradient $\tau_G$, Loss $\tau_L$, Fisher Information $\tau_{FI}$, Energy $\tau_E$, Uncertainty $\tau_U$, and Features $\tau_F$ (see full notations in Table 15) in predicting novelty. We provide the numerical value of MStats, and the ones computed by Z-norm $f_z(\cdot)$. We report the value of a 'Novelty' metric from 1 step prior to a novel task $\rightarrow$ when a novel task appears. The linear separability can be evaluated by the numerical difference between values where $f_z$ provides a larger difference (*e.g.* $-0.02 \rightarrow 4.27$ compared to $0.16 \rightarrow 0.16$) where an MStats can then be utilized for novelty prediction. Linear separability can also be used to evaluate the informativeness of a MStats in predicting novelty, where higher separability would indicate a greater discriminative power for a MStats. We find that $f_z(\|\overline{\tau_F}\|_2)$ and $f_z(\tau_L)$ perform the best among other MStats. We observe that the results correlate with the evaluation of the same MStats in novelty prediction in Table 16.

| Novelty | Cross-Modal | Permutation | In-Modal | Distribution |
|---|---|---|---|---|
| $f_z(\|\tau_G\|_2)$ | $-0.01 \rightarrow 11.74$ | $-0.04 \rightarrow 10.08$ | $-0.08 \rightarrow 6.58$ | $-0.06 \rightarrow 7.29$ |
| $\|\tau_G\|_2$ | $0.09 \rightarrow 0.41$ | $0.09 \rightarrow 0.37$ | $0.09 \rightarrow 0.30$ | $0.09 \rightarrow 0.28$ |
| $f_z(\tau_L)$ | $-0.06 \rightarrow 34.86$ | $-0.09 \rightarrow 27.02$ | $-0.18 \rightarrow 15.38$ | $-0.02 \rightarrow 19.17$ |
| $\tau_L$ | $1.08 \rightarrow 5.25$ | $1.07 \rightarrow 4.41$ | $1.05 \rightarrow 3.14$ | $1.04 \rightarrow 3.37$ |
| $f_z(\overline{\tau_{FI}})$ | $-0.02 \rightarrow 4.27$ | $-0.02 \rightarrow 2.92$ | $0.02 \rightarrow 0.83$ | $-0.02 \rightarrow 1.71$ |
| $\overline{\tau_{FI}}$ | $0.16 \rightarrow 0.16$ | $0.16 \rightarrow 0.17$ | $0.16 \rightarrow 0.17$ | $0.16 \rightarrow 0.17$ |
| $f_z(\tau_E)$ | $-0.02 \rightarrow 2.99$ | $-0.07 \rightarrow 2.88$ | $0.003 \rightarrow 2.46$ | $-0.06 \rightarrow 2.46$ |
| $\tau_E$ | $-5.48 \rightarrow -5.04$ | $-5.43 \rightarrow -4.98$ | $-5.31 \rightarrow -4.88$ | $-5.35 \rightarrow -5.00$ |
| $f_z(\overline{\tau_U})$ | $0.01 \rightarrow 1.93$ | $-0.04 \rightarrow 1.78$ | $-0.01 \rightarrow 1.53$ | $0.04 \rightarrow 1.51$ |
| $\overline{\tau_U}$ | $0.11 \rightarrow 0.13$ | $0.11 \rightarrow 0.13$ | $0.11 \rightarrow 0.12$ | $0.10 \rightarrow 0.12$ |
| $f_z(\|\overline{\tau_F}\|_2)$ | $0.003 \rightarrow 35.91$ | $0.03 \rightarrow 18.95$ | $-0.08 \rightarrow -2.48$ | $0.05 \rightarrow -2.26$ |
| $\|\overline{\tau_F}\|_2$ | $184.19 \rightarrow 142.07$ | $165.70 \rightarrow 148.18$ | $141.75 \rightarrow 132.37$ | $142.21 \rightarrow 137.83$ |

threshold on the output of a one-step Frank-Wolfe algorithm applied on the uncertainty (entropy) value of the learner; $f_{\text{OneShot}}(\tau_U)$. The methods described use a fixed threshold that is based on a heuristic to infer novelty. We use them as baselines to compare with our method $\alpha$MetaSup, which is adaptive and infers novelty using a neural network. In addition, we extend an OOD detection method in an online fashion where it applies a threshold using the energy score (Liu et al., 2020) $f_E(\tau_E)$ as a baseline in our experiments.

**Experience Replay** (ER) (Bagus & Gepperth, 2021) randomly and uniformly sample (Rolnick et al., 2019) from all training data that are stored in a limited-size buffer that is rehearsed during training. During rehearsal, the buffer data are appended to the current training batch and used as an auxiliary to the current training objective. Gdumb (Prabhu et al., 2020) questions the importance of training on the full task in a CL setting. During training time, they do not train a model but instead randomly sample data to store in a buffer, during test time they train a new model on the stored buffer data. Extensions (Aljundi et al., 2019b; Arani et al., 2022; Bang et al., 2021; Buzzega et al., 2020b) use a heuristic on 'what to sample' from a given batch, such as the most informative samples. While fewer work (Sun et al., 2022) provide improvements on the timestep of storing samples to memory ('when to sample'). 'When to sample' can be based on criteria such as a novel task while 'what to sample' can be used to decide which samples to store and replace from memory. As such, the two methods can be used in combination, where the learner's performance can be improved by efficient memory utilization. Our work introduces a method of 'when to sample' with the use of $\alpha$MetaSup and Surprise Sampling. We evaluate the efficiency of memory utilization our method provides to show an improvement over current baselines. Extensions to ER such as (Buzzega et al., 2020a; Arani et al., 2022) do not rely on the task identity to perform GCL. Such and similar work can be improved with methods of 'when to sample' as opposed to uniform random sampling from all training batches. Our *Meta Sampling* ('MSamp') method can be applied in extension to ER learning algorithms and collect samples prior to a novel task transition (*surprise*). Our findings in Section 6.3 of the main text agree with (Sun et al., 2022) that uniform sampling under-performs compared to MSamp and when evaluated at GCL.

## C METHODS

In this section, we provide a detailed description of the methods used in the main text to ablate the performance of $\alpha$MetaSup and evaluate Stream.

### C.1 META-STATISTICS

Meta-Statistics (MStats) are computed during a learner's training and are used online to analyze the learning process. We use MStats as a method to quantify the novelty level of a new task. We group the MStats listed in Table 2 of the main text by their dimensionality. At a training step $i$, Loss, Uncertainty, and Energy MStats are scalars $\tau_L, \tau_U, \tau_E \in \mathbb{R}$; While Gradient, Fisher Information, and Feature MStats $\tau_G, \tau_{FI}, \tau_F$ are lists of $|\theta|$ tensors where $|\theta|$ is the number of layers and each tensor $\tau_i$ in the list has dimensions $\tau_i \in \mathbb{R}^{b \times I_i \times O_i}$ that depend on batch size $b$, input $I_i$ and output $O_i$ dimensions. As such, a **dimensionality reduction** method is required for using $\tau_G, \tau_{FI}, \tau_F$ for the novelty detection task. Previous works perform *dimensionality reduction* to aggregate them as an intermediate step before using them with their method. For example, for an 'n' layer Feed-Forward Network the Gradient MStats would be a list $\tau_G = \{\tau_1, \ldots, \tau_n\}$ where the dimension of $\tau_i \in \mathbb{R}^{b \times I_i \times O_i}$ would depend on the batch size $b$, the input dimension to layer $i$, $I$ and the output dimension $O$. For example, $\tau_G$ can be reduced by the L2 norm, $||\tau_G||_2 = \{||\tau_1||_2, \ldots, ||\tau_n||_2\}$, where $||\tau_G||_2$ would be a list of scalars.

We find that directly using the high-dimensional representation is uninformative. Our work is the first to perform a systematic evaluation of MStats and the reduction methods that make them utilizable for novelty identification. We apply dimensionality reduction on each tensor in the list, *e.g.* the reduced MStats will be a list of scalars where each scalar quantifies the novelty in that layer. We experiment with two reduction methods, *mean* and *L2 norm*, in our search for the best indicator of novelty level, and present the result together with heuristic novelty calculation in Table 16, Table 17, Table 18, and Table 19.

### C.2 NOVELTY DETECTION METHODS

We identify and adapt several methods used in literature for identifying novelty and design 3 novel methods based on the Running Mean $f_z$ (Aljundi et al., 2019a), Contrastive similarity $f_{sim}$ (Aljundi et al., 2019b), and Bayesian approaches $f_{bayes}$ (Kirichenko et al., 2021), to evaluate the surveyed MStats.

For an MStats $\tau$ we use $f_h$ to map $\tau_i$ from train step $i$ to a scalar that quantifies novelty. Thereby we find $\eta$, s.t. maximize the F1 score on the dummy Stream:

$$\max_{\eta} F1(f_h(\tau_i) > \eta : i \to \texttt{Surprise}) \tag{2}$$

**Running Mean** $f_z$ maintains statistics from the meta-learning process Meta-Statistics $\tau$ as learnable parameters, running mean $\bar{\mu}_\tau$ and running std $\bar{\sigma}_\tau$. Similar to Batch Normalization (Ioffe & Szegedy, 2015), the moving average $\bar{\mu}_\tau$ and standard deviation $\bar{\sigma}_\tau$ are updated with momentum $\lambda$ with MStats $\tau_i$ computed from the data batch at each training step $i$. Since $\tau_i$ can be a vector, we use an aggregation method $\texttt{Agg}$ (*e.g.* mean, max) to compute a scalar value as the degree of novelty.

$$
\begin{aligned}
f_z(\tau_i) &= \texttt{Agg}[(\tau_i - \bar{\mu}_\tau)/\sqrt{\bar{\sigma}_\tau + \epsilon}\,] \\
\bar{\sigma}_\tau &\leftarrow \bar{\mu}_\tau + \lambda(\tau_i - \bar{\mu}_\tau) \\
\bar{\mu}_\tau &\leftarrow (1 - \lambda)\big[\bar{\sigma}_\tau + \lambda(\tau_i - \bar{\mu}_\tau)^2\big]
\end{aligned}
\tag{3}
$$

**Contrastive Similarity** $f_{sim}$ has been used to select rehearsal samples for diversity (Aljundi et al., 2019b) and mitigate forgetting. We extend the approach to identify novelty, by contrasting MStats from the current training step $\tau_i$ with a reference window of MStats from the previous training steps $\mathcal{W} = \{\tau_{i-w}, \ldots, \tau_{i-1}\}$. We compute $f_{sim}$ as the average cosine *dissimilarity* from $\tau_i$ to all MStats in $\mathcal{W}$, where a higher value corresponds to a higher novelty level:

$$f_{sim} = \frac{1}{|\mathcal{W}|} \sum_{j=i-w}^{i-1} \frac{\tau_i \cdot \tau_j}{||\tau_i||\,||\tau_j||} \tag{4}$$

**Bayesian** $f_{\textbf{bayes}}$ maintains the mean $\bar{\mu}_\tau$ and the log std $\bar{\sigma}_\tau$ from $f_z$ and similarly to $f_{\text{sim}}$ maintains a window of MStats $\mathcal{W}$. Novelty is computed on the KL divergence where the prior is the training moving average, and the posterior distribution is the batch average. The KL divergence between the two distributions $\mathcal{N}(\bar{\mu}_\tau, \bar{\sigma}_\tau)$ and $\mathcal{N}(\bar{\mu}_\mathcal{W}, \bar{\sigma}_\mathcal{W})$ provide the degree of novelty:

$$f_{\text{bayes}} = \sum [\bar{\sigma}_\tau - \bar{\sigma}_\mathcal{W} + \frac{\exp(\bar{\sigma}_\mathcal{W})^2 + (\bar{\mu}_\tau - \bar{\mu}_\mathcal{W})^2}{2\exp(\bar{\sigma}_\tau)^2} - 0.5] \tag{5}$$

Where higher dissimilarity between batch statistics and running statistics would result in higher KL-Divergence and thus a higher degree of novelty.

Our **Meta-Learning Process** first collects MStats data from a dummy learner trained on a dummy Stream task sequence that does not deploy a mechanism to mitigate forgetting. In the second stage, we tune or train a method such as $f_{\text{h}}$ (heuristic) or $f_\alpha$ (adaptive) on the data collected from the first step. Finally, we use the learned $\eta$ or $f_\alpha$ from the previous step with a new learner of the same architecture and a method that mitigates forgetting to learn a novel task sequence (GCL task sequence), which consists of different tasks agnostic of the dummy Stream.

For all of our experiments, we collect MStats from 100 dummy learners trained on 100 tasks of S-Num, S-Modal, and S-Vis. We use $\lambda = 0.01$ for $f_{\text{z}}$ and $f_{\text{bayes}}$.

### C.3 $\alpha$METASUP

Our meta-learning process using Stream allows us to 'learn' $\eta$, but novelty detection using a threshold cannot fully address the issue of non-uniform learning-gaps. Motivated by the shortcomings of a fixed threshold, we propose a baseline method to Stream, $\alpha$MetaSup.

$\alpha$MetaSup uses a neural network on a window of MStats that predicts a novel task, whereas the aforementioned methods compute the novelty score based on the current training batch. Training a neural network can require a significant amount of data where novel tasks can be sparse. Stream provides a method to construct large task sequences with sufficient true-positives that can be used to observe the behavior of a dummy learner in a diverse set of circumstances. The advantage of the diversity of task sequence provided by Stream is reflected in the improved generalization of $\alpha$MetaSup between two Stream sequences constructed using different datasets as opposed to standardized benchmark sequences (*e.g.* PMNIST).

$\alpha$**MetaSup Meta-Learning Process.** Similar to the meta-learning process of Appendix C.2 we collect MStats which we use as a dataset to train a small neural network $f_\alpha$. We perform novelty detection as a time series classification problem and use a sequence of length $w$ as inputs to the neural network. We annotate each sequence as a positive sample if there is a task transition within the 5 most recent training steps, and a negative sample otherwise. We use $w = 10$ for all $\alpha$MetaSup experiments.

Due to the low positive rate of novel task transition steps, the output probabilities of the trained $f_\alpha$ are miscalibrated. We calibrate $f_\alpha$ after training using isotonic regression (Guo et al., 2017) on the same MStats dataset. $\alpha$MetaSup is used in an online manner during GCL to infer novelty.

**Classifier Architecture.** We evaluate two classifier choices: A unidirectional LSTM $f_{\text{lstm}}$ where the classification head is attached to the final hidden state, and a Transformer $f_{\text{gpt}}$ where the input is processed by one embedding layer, one positional embedding layer, one multi-head self-attention layer, and finally the binary classification head. We perform an ablation study in Appendix E.4 to evaluate the generalization and robustness of each model.

## D  META-SAMPLING

Rehearsal Learning methods sample uniformly over all training steps (*e.g.* DER++ (Buzzega et al., 2020a)) and are effective when task identity is absent. Similarly to our work, recent work (Sun et al., 2022) has found that oversampling ultimately degrades performance, where samples from memory can be replaced at random. We found in the main text Sec. 5.3 that current state-of-the-art rehearsal-based methods cannot perform on our challenging Stream S-Modal. To provide baseline results and aid future research, we propose Meta-Sampling for these methods to generalize on Stream.

---

**Algorithm 1** Meta-Sampling

---

**Given:** dataset $\mathcal{D}$, novelty detection method $f_\alpha$, Meta-Statistics collection function $f_\tau$, CL rehearsal method $\mathcal{L}_{\text{CL}}$.
**Initialize:** learner $f_\theta$, short-term memory $\mathcal{B}$, long-term memory $\mathcal{M}$, warm-up steps $k$.
**for** $x_i, y_i$ in $\mathcal{D}$ **do**
    $\tau_i = f_\tau(f_\theta(x_i), y_i)$
    **if** $f_\alpha(\tau_{i-w:i})$ and $k = 0$ **then** {*Novelty Detected*}
        $B \leftarrow \texttt{Sample}(\mathcal{B})$
        $\mathcal{M} \leftarrow B \cup \mathcal{M}$
        reset $\mathcal{B}, k$

    Compute task loss: $\ell = \mathcal{L}(f_\theta(x_i), y_i)$.
    Compute CL loss: $\ell_{CL} = \mathcal{L}_{CL}(\mathcal{M})$
    Update learner: $f_\theta \leftarrow \nabla(\ell + \ell_{CL})$

    **if** $k = 0$ **then** $\mathcal{B} \xleftarrow{\textbf{Append}} (x_i, y_i)$
    **else** $k \leftarrow k - 1$ {*Warm-up Phase*}
**end for**

---

*Meta-Sampling* (MSamp) selects a number of recent samples $B$ from the short-term memory $B \subset \mathcal{B}$ to store in the long-term memory $\mathcal{M}$. $B$ include samples directly **prior** to a novel task transition ('surprise') and not including the current training batch classified as novel. Surprise Sampling can work with any method that can perform Novelty Detection and be used by any Continual Learning method that utilizes a memory of some sort. We use **warm-up** to avoid novelty detection during the learning phase of a new task. The algorithm is provided in Algorithm 1. MSamp can be used to identify training batches from which to sample, while additional methods such as (Aljundi et al., 2019b) can be used to select which samples to replace from memory from the current time step. We compare our method with current methods that sample every batch uniformly as a baseline.

We find several similarities between Meta-Sampling and the biological mechanism of learning in mammals. It has been known that experiences that promote adrenal arousal lead to the movement of recent experiences to long-term memory in humans (McGaugh, 2013). Additionally, a learner can be less sensitive to novelty after a surprising event and requires time to acclimate. Last, surprise can be induced by both traumatic and novel experiences, but storing them can often be counterproductive to a learner. When novelty is detected, MSamp provides a mechanism to 'solidify' prior experiences. Similarly, events leading to 'surprise' can be important for learning causal relationships.

Both our insight and our approach are contrary to the methods (Aljundi et al., 2019b;a) that sample **during** novelty detection which we found to perform poorly. We find that novelty can be triggered by more than novel data, such as noisy data, which would lead to a false-positive response. Current CL methods are not robust to false positives; such as applying a CL method to noisy data leads to catastrophic forgetting.

# E  ADDITIONAL EXPERIMENTS

## E.1  LEARNING-GAPS

We provide a full table of learning-gap ablation study mentioned in Section 5.1 in Table 7. The strength of transformation is defined by the number of degrees in the Rotation Transformations. The rotation degree greatly affects the magnitude of In-Modal and Distribution Gaps, and has little effect on Cross-Modal Gaps. Permutation Gap is similar to Cross-Modal Gap in magnitude and therefore we group these learning-gaps into three categories (Cross and In-Modal, Distribution) in Figure 2. The results agree with our design of the Stream Transformations.

Table 7: We evaluate learning-gap categories identified in Figure 2 for S-Modal. We report the learner's Forward Transfer (FWT) (Lopez-Paz & Ranzato, 2017) on an unobserved task $t_b$ after being trained on task $t_a$ as the magnitude of the learning-gap $t_a \rightarrow t_b$. We observe that Permutation transformations, create distinct tasks where the performance is close to random (AUC≈0.5). We apply Rotation to the same dataset permutation ('Distribution') and observe a correlation between the degrees and the learning-gap. Similarly for transitions between tasks of the same modality ('In-Modal'), but with different base datasets and degrees. Last, task transitions between different modalities and degrees ('Cross-Modal') have no FWT between the tasks, implying low task similarity. Our results justify the choice of transformations and showcase the diversity of task transitions provided by Stream. '-' denotes inapplicable, where Permutation Gap is invariant w.r.t. rotation degrees, and Distribution Gap by 0 degree implies $t_a = t_b$ (no learning-gap).

| Degrees $T_{\text{rot}}$ | FWT - AUC | | | |
|---|---|---|---|---|
| | Permutation | Cross-Modal | In-Modal | Distribution |
| 0 | 0.50 | 0.51 | 0.76 | - |
| 5 | - | 0.49 | 0.56 | 0.57 |
| 10 | - | 0.49 | 0.52 | 0.51 |
| 15 | - | 0.49 | 0.50 | 0.50 |
| 20 | - | 0.49 | 0.50 | 0.48 |
| 25 | - | 0.50 | 0.49 | 0.47 |
| 30 | - | 0.50 | 0.52 | 0.48 |

Table 8: Evaluation of task difficulties when using different Concept Mappings to group 345 DomainNet classes into 10 concepts. Our concept mapping increases the AUC by 1.07% and 6.34% for 'Real' and 'Sketch' datasets respectively, which has a lower difficulty compared to random mapping (*i.e.* a 2.4% theoretical upper-bound increase by our mapping). Although random mapping increases the difficulty, it is not catastrophic, as the learner could still learn the tasks using the concepts. We conclude that learning-by-concepts is feasible in our setting and that the class-to-concept mapping must be identical for an equivalent comparison between methods. We provide our mapping in Table 13 and Table 14.

| Dataset | Concept Mapping | AUC | Std |
|---|---|---|---|
| Real | Ours | **0.980** | 0.001 |
| | Random | 0.969 | 0.001 |
| Sketch | Ours | **0.908** | 0.002 |
| | Random | 0.851 | 0.004 |
| S-Modal (Upper-Bound) | Ours | **0.812** | - |
| S-Modal (Upper-Bound) | Random | 0.793 | - |

### E.2 STREAM CONCEPT MAPPING

We evaluate the impact on the performance of the method used to map classes into concepts. For our experiments, we assign 345 DomainNet classes to 10 concepts for Real and Sketch-based tasks in S-Modal. We use K-Means clustering on the embedding space of a CLIP model that gives the textual description of each class. We compare our method of mapping classes to concepts to randomly assigning classes to concepts. For each of $D_{\text{base}} \in \{\text{Real}, \text{Sketch}\}$, we report the task difficulty in AUC averaged from 20 seeds of 100 transformations when training each transformed task separately without CL. In Table 8, we conclude that using random concept mapping does not lead to inability to learn a task, where learning is still possible although more difficult. As such, our concept-mapping method has merit, where the difficulty of learning a task decreases. Our findings suggest that the concept-mapping method is orthogonal to the evaluation of the learner but must be identical when evaluating between learning algorithms. We provide our concept mapping for Real and Sketch in Table 13 and Table 14.

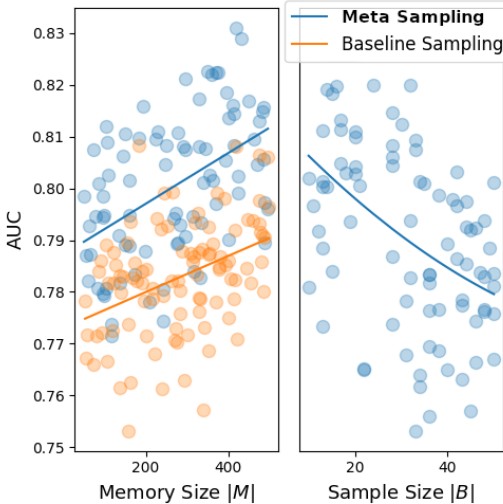

Figure 4: Ablation study on the efficiency when constructing a memory using Meta-Sampling (blue markers) and Baseline Sampling (Reservoir (Rolnick et al., 2019), orange markers). Each marker corresponds to the performance of a learner that uses Experience Replay augmented by the two aforementioned sampling methods. We evaluate the learners on 20 S-Num tasks trained and for 100 random repetitions. **Left**, our method outperforms the baseline where the blue line has a larger y-intercept. Additionally, the performance improves with the Memory Size at an increased rate; as such, our method is memory-efficient. **Right**, in Meta-Sampling, a larger Sample Size $|B|$ degrades performance as a larger part of $M$ is replaced with recent data and increases the in-memory task imbalance. We conclude that efficient memory construction is an important element in Continual Learning.

### E.3    META-SAMPLING

We examine the efficiency of our method in constructing a memory $\mathcal{M}$ when using our method, Surprise Sampling, and compare to Reservoir Sampling (Rolnick et al., 2019) (uniform). We train 100 models on S-Num and report the test-set mean AUC at the end of 20 tasks. Surprise Sampling is more memory-efficient where the improvement in AUC is at an increased rate Figure 4. Our experiments confirm our hypothesis from the main text Section 6.3, where sampling prior to a task transition is an improvement compared to sampling from all batches. Additionally, we identify degradation as we increase the size of the temporary buffer where a larger size is more likely to replace older memories with recent data and increases the imbalance, which harms the CL method that will utilize the memory.

Our results agree with previous work (Ardywibowo et al., 2022) which found that accurate inference of a novel task can improve the performance of a CL method under general conditions. However, we also observe the contrary, where $f_\alpha$ with a high false-positive rate used with Surprise Sampling and a small $|B|$ can out-perform a low false-positive $f_\alpha$ when used in conjunction with SS with an equivalent $|B|$. We conclude that the robustness of the CL method is important where not applying a CL method during a novel task is often more beneficial than applying a CL method to the wrong task.

### E.4    $\alpha$METASUP

We evaluate the generalization of an adaptive novelty detection method $\alpha$MetaSup ('$f_{\text{gpt}}$') compared to a heuristic method ('$f_z$') to supplement our experiments from Section 6.2 in the main text. We use 100 S-Num task sequences to observe a ResNet-18 and Residual MLP dummy learner with a novelty detection method. After we tune the novelty detection method to the dummy learner, we use it with a new learner on a new task sequence. Our experimental setting evaluates the most effective MStats to use as well as the novelty detection method that generalizes the best among the backbones. In Table 9, $\alpha$MetaSup outperforms the heuristic methods by a significant margin, and $f_{\text{gpt}}$ achieves

Table 9: We evaluate how a novelty detection method generalizes when trained with our meta-learning process from Appendix C.2. We use S-Num with both ResMLP and ResNet-18 dummy learners to train a novelty detection method, $f$. We evaluate $f$ on a new learner when training on tasks S-Modal and S-Vis respectively. We report the novelty detection F1 scores where train $\rightarrow$ validation performance indicates the generalization of $f$. $\alpha$MetaSup $f_{\text{gpt}}$ achieves the highest F1 during meta-learning under a more challenging setting with a 'ResNet-18' dummy learner. We conclude that $f_{\text{gpt}}$ has an improved generalization over heuristic detection methods $f_z$.

| Novelty Detection Learner | S-Num $\rightarrow$ S-Vis ResNet-18 | S-Num $\rightarrow$ S-Modal Residual MLP |
|---|---|---|
| $f_z(\tau_L)$ | $0.960 \rightarrow 0.694$ | $0.778 \rightarrow \mathbf{0.932}$ |
| $f_z(\|\tau_G\|_2)$ | $0.929 \rightarrow 0.726$ | $0.663 \rightarrow 0.749$ |
| $f_z(\overline{\tau_{FI}})$ | $0.560 \rightarrow 0.356$ | $0.434 \rightarrow 0.102$ |
| $f_z(\overline{\tau_E})$ | $0.945 \rightarrow 0.475$ | $0.636 \rightarrow 0.437$ |
| $f_z(\overline{\tau_U})$ | $0.680 \rightarrow 0.025$ | $0.321 \rightarrow 0.079$ |
| $f_z(\|\overline{\tau_F}\|_2)$ | $0.628 \rightarrow 0.455$ | $0.380 \rightarrow 0.838$ |
| $f_{\text{lstm}}(\|\tau_G\|_2)$ | $0.991 \rightarrow 0.768$ | $0.815 \rightarrow 0.862$ |
| $f_{\text{gpt}}(\|\tau_G\|_2)$ | $\mathbf{0.991} \rightarrow \mathbf{0.908}$ | $\mathbf{0.822} \rightarrow 0.814$ |

the highest F1 for most protocols. When $f_{\text{gpt}}$ and $f_{\text{lstm}}$ are evaluated by the in-train performance to S-Vis and S-Modal they perform with average F1 scores of 0.861 and 0.815 respectively, and both are better than $f_z$ (0.813). We conclude that $f_{\text{gpt}}$ is a better design choice over $f_{\text{lstm}}$ for $\alpha$MetaSup. Interestingly, we find a counterintuitive result where some MStats and $f_z$ generalize better in the validation set for simpler settings (*e.g.* using a Residual MLP compared to ResNet-18). We find this to be an artifact of the low true-positive rate where an optimistic threshold (*e.g.* larger value) appears to perform better. We hypothesize that improvements in the architecture of $f_{\text{gpt}}$ can further reduce the generalization gap in S-Modal.

Table 10: Relative training time efficiency when evaluating GCL methods on S-Modal. We evaluate the relative training time of our novelty detection method at S-Modal and compare it to a 'Dummy' learner with no CL method (fine-tuning). We compare the time cost of a GCL method in the top half of the table to the Meta-Sampling augmented ER (we denote as 'S-ER') that infers novelty using one of the baselines to apply Meta-Sampling. We find that Meta-Sampling reduces the time cost by half. Next, we evaluate the time efficiency of $\alpha$MetaSup and find $f_{\text{gpt}}$ performs more efficiently with only 106.5% relative difference to a 'Dummy' learner. Our novelty detection method induces no extra time cost despite using a neural network, while Surprise Sampling effectively reduces the time cost of the augmented CL method.

| Method | Novelty Detection | Relative Time |
|---|---|---|
| Dummy | - | 100.0% |
| ER | - | 228.1% |
| DER++ | - | 229.0% |
| CLS-ER | - | 214.7% |
| S-ER | $f_{\text{Peak}}(\tau_L)$ | 108.7% |
| | $f_{\text{OneShot}}(\tau_U)$ | 113.1% |
| | $f_{\text{EWMA}}(\tau_L)$ | 113.1% |
| | $f_E(\tau_E)$ | 110.7% |
| **S-ER** | $f_{\mathbf{gpt}}(\|\tau_G\|_2)$ | **106.5%** |

**Time Efficiency Evaluation.** We evaluate the time of S-Modal training of $\alpha$MetaSup augmented Surprise Sampling (SER) compared to uniform sampling baselines and previous heuristic methods (Aljundi et al., 2019a; Wortsman et al., 2020; Zhu et al., 2022; Liu et al., 2020). In Table 10, $\alpha$MetaSup Surprise Sampling has the highest time efficiency, which is the closest to a 'Dummy' learner.

## E.5   MULTI-MODAL STREAM

We supplement the training curve for the S-Modal benchmark experiment in Section 6.3 in the main text.

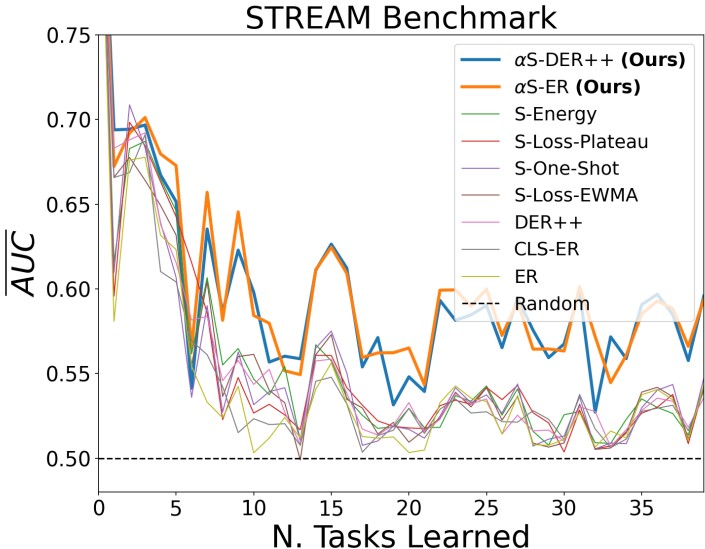

Figure 5: We visualize the mean AUC as we progressively learn more tasks. The variance in Mean AUC reflects the changing difficulty of tasks over time, such as an easy task followed by more difficult ones. All methods are trained and evaluated on an identical sequence of S-Modal tasks and for 252 random initializations. $\alpha$MetaSup augmented methods are the only ones that do not have their performance degraded close to Random.

# F   EXPERIMENTAL SETUP

We perform all our experiments on a cluster of 8 x V100 GPUs. We use Stream and construct sequences of tasks with *abrupt* transitions, *e.g.* each batch contains samples from the same task. For our transformations we randomly sample the angle from $\{30, 60, \ldots, 330\}$ for images and $\{5, 10, \ldots, 40\}$ for feature vectors. We construct permutation transformations by sampling a random seed and applying the same transformation to all base-dataset and only once for each task sequence. Finally, we shuffle the task sequence, where we keep a fixed random seed of 0 for all the previous steps. **Meta-Statistics Evaluation Setup.** For evaluation of MStats from Section 6.2 in the main text and Appendix E.4, we use a Dummy learner to collect statistics on 100 task sequences for each of PMNIST, Split-CIFAR100, S-Num, and S-Vis task sequences. We use 5 permutations and 5 rotations for each task sequence and, therefore, there are 20 tasks in S-Num and S-Vis. We use the same collected statistics for the meta-learning process of all novelty functions. The training steps that belong to task transitions are naturally imbalanced when compared to the training steps of the task sequence. We evaluate the novelty detection performance using the F1 score and set it as true-positive when any positive response appears within the window of 5 batches after a task transition.

**$\alpha$MetaSup Training Setup.** We train $\alpha$MetaSup as a time series classification problem and discuss details in Appendix C.3. We train the NN classifiers for 50 epochs using a learning rate of 0.1 and an SGD optimizer with a momentum of 0.5.

**Stream Benchmark Experiment Setup.** For image datasets (*e.g.* 'S-Vis'), we use a randomly initialized ResNet-18 backbone, while for featurized vector datasets (*e.g.* 'S-Modal') we train a two-layer Residual Multi-Layer-Perception ('ResMLP'). For our experiments that use ResNet-18, we replace and freeze the first convolutional layer as our feature extraction method to make the backbone applicable to our setting and as discussed in Appendix A.

We train for a minimum of 2 epochs for S-Num and a minimum of 4 epochs for S-Vis and S-Modal. After we reach the minimum number of epochs, we switch to a new task with a random probability of $p = 0.01$ evaluated at each training batch. We use an initial learning rate of 0.1 and an SGD optimizer with a momentum of 0.5 to train all datasets. Hyper-parameters of our method are provided in Table 11.

In the Stream Benchmark experiment, we compare the performance of the GCL baselines ER (Riemer et al., 2018), DER++ (Buzzega et al., 2020a) and CLS-ER (Arani et al., 2022), to the same method ER but where we augment with Meta-Sampling and a novelty detection method. For novelty detection baselines we compare $\alpha$MetaSup with $f_{\text{Peak}}(\tau_L)$ (Aljundi et al., 2019a), $f_{\text{OneShot}}(\tau_U)$ (Wortsman et al., 2020), $\tau_E$ (Liu et al., 2020) and $f_{\text{EWMA}}(\tau_L)$ (Zhu et al., 2022). For all novelty detection baselines, we use the reported threshold values as an initial guess and iteratively tune the method to find a threshold that improves novelty detection performance. Our method of tuning is identical to current practices for finding the threshold (*e.g.* ad-hoc). Hyper-parameters for GCL baselines, as well as novelty detection baselines, are listed in Table 12. We evaluate across datasets where we use the threshold we find in S-Num and apply it in S-Vis and S-Modal.

Table 11: $f_h$, $\alpha$MetaSup and Meta-Sampling hyper-parameters. Explanations of the hyper-parameters are provided in Appendix C.2 ($f_h$), Appendix C.3 ($\alpha$MetaSup), and Appendix D (Meta-Sampling).

|  | Hyper-parameter | Value |
|---|---|---|
| $f_z, f_{\text{bayes}}$ | Momentum $\lambda$ | 0.01 |
| $f_{\text{sim}}, f_{\text{bayes}}$ | Reference window size $|\mathcal{W}|$ | 10 |
| $\alpha$MetaSup | MStats sequence length $w$ | 10 |
|  | Hidden dim | 200 |
|  | Dropout | 0.3 |
| Meta-Sampling | Short-term buffer size $|\mathcal{B}|$ | 100 |
|  | Sampling Size $|B|$ | 10 |

Table 12: Stream benchmark experiment hyper-parameters. **GCL Baselines**: ER (Riemer et al., 2018), DER++ (Buzzega et al., 2020a) and CLS-ER (Arani et al., 2022) randomly retrieve from the memory at each training step replaying past experiences, where 'Replay size' is the number of samples retrieved during rehearsal step of the CL method. Where applicable, we use the values reported in the original paper of each method. **Novelty Detection Baselines**: $f_{\text{Peak}}(\tau_L)$ (Aljundi et al., 2019a), $f_{\text{OneShot}}(\tau_U)$ (Wortsman et al., 2020), $\tau_E$ (Liu et al., 2020) and $f_{\text{EWMA}}(\tau_L)$ (Zhu et al., 2022) use a fixed thershold to trigger novelty. We iteratively tune the threshold specific to each method to improve novelty detection performance on Dummy learners (*e.g.* ad-hoc). For all other hyper-parameters we use the ones reported in the original paper.

|  | Hyper-parameter | Values |
|---|---|---|
| Training | Batch size | 64 |
|  | Memory size $|\mathcal{M}|$ | 500 |
|  | Replay size | 128 |
| **GCL Baselines** |  |  |
| ER | Task loss coef. | 1.0 |
| DER++ | Task loss coef. | 1.0 |
|  | Distill coef. | 1.0 |
| CLS-ER | Task loss coef. | 1.0 |
|  | Regularization coef. | 1.0 |
|  | Plastic model update frequency | 1.0 |
|  | Plastic model alpha | 0.99 |
|  | Stable model update frequency | 0.9 |
|  | Stable model alpha | 0.99 |
| **Novelty Detection Baselines** $f_h(\tau)$ |  |  |
| $f_{\text{Peak}}(\tau_L)$ (Aljundi et al., 2019a) | Plateau window size | 5 |
|  | Loss smooth steps | 5 |
|  | Loss peak threshold | 1.0 |
|  | Loss plateau threshold | 0.1 |
| $f_{\text{OneShot}}(\tau_U)(Wortsman\ et\ al.,\ 2020)$ | Threshold | 1.5 |
| $\tau_E$ (Liu et al., 2020) | Threshold | -3.0 |
|  | Temperature | 1.0 |
| $f_{\text{EWMA}}(\tau_L)(Zhu\ et\ al.,\ 2022)$ | Threshold | 2.0 |
|  | EWMA window size | 100 |
|  | Smooth factor | 0.2 |

Table 13: We divide the table into two parts with the second part presented in Table 14. The concepts are created by performing K-Means clustering on the features extracted using a CLIP (Radford et al., 2021a) model from the textual description of a class. 345 DomainNet classes are mapped to 10 concepts. A smaller number of concepts is limited to create a semantically meaningful mapping of the 345 classes, where for example, 'sun' and 'hamburger' are mapped to the same Concept ('Outdoor and Wildlife'). We justify that fuzzy mapping can influence task difficulty but does not make learning prohibitive as we find by our experiments in Table 8. As we evaluate the learning algorithm's performance, an identical mapping must be used when evaluating between methods. A method to make a semantically meaningful mapping can be a research problem of its own (Liu et al., 2007; Cai et al., 2022). We conclude that learning concepts in a Domain-Incremental Learning (DIL) setting is a viable option to evaluate the performance of the learning algorithm.

| S-Modal Concept | DomainNet Classes |
| --- | --- |
| Food | strawberry, cookie, coffee cup, lollipop, sandwich, ice cream, hot dog, toothpaste, donut, cake, birthday cake, watermelon, pizza, steak, bread, popsicle |
| Outdoor and Wildlife | saw, eraser, arm, camera, dog, tornado, ear, radio, panda, pliers, leg, piano, bed, hexagon, potato, crocodile, rollerskates, computer, pig, parachute, palm tree, flamingo, key, map, anvil, cat, penguin, elephant, campfire, fan, zigzag, vase, sun, kangaroo, axe, helmet, hedgehog, rhinoceros, yoga, umbrella, tiger, hamburger, bee, camel, octopus, cactus, bus, banana, zebra, van, snorkel, harp, ant, helicopter, car, rifle, owl, pencil, calendar, cow, bat, hat, guitar, cup, broccoli, skull, ocean, hammer, asparagus, angel, peanut, marker, camouflage, eye, toe, telephone |
| City Objects | the eiffel tower, flashlight, hot air balloon, fire hydrant, wine glass, hourglass, dumbbell, house plant, stairs, binoculars, megaphone, skyscraper, streetlight, candle, stop sign, wine bottle, eyeglasses, stethoscope, windmill, bulldozer, picture frame, ceiling fan, traffic light, lighthouse, floor lamp, the great wall of china, light bulb, necklace, chandelier, microphone, fireplace, lantern |
| Household | hot tub, bridge, fence, jacket, diving board, pond, postcard, table, toilet, pool, garden, sink, bathtub, octagon |
| Body | finger, foot, pillow, bowtie, bandage, shoe, wristwatch, beard, pants, t-shirt, tooth, stitches, elbow, bracelet, belt, shorts, lipstick, underwear, sweater, moustache, mouth, knee, goatee, sock, flip flops, string bean, mug, hand |

Table 14: The table is used auxiliary to Table 13.

| S-Modal Concept | DomainNet Classes |
| --- | --- |
| Animals & Plants | shark, peas, blackberry, mermaid, butterfly, rabbit, monkey, sheep, sea turtle, sea turtle, raccoon, pineapple, grapes, bird, feather, bear, lion, dragon, crab, giraffe, carrot, frog, swan, animal migration, teddy-bear, mushroom, spider, fish, lobster, whale, mouse, scorpion, onion, snail, parrot, squirrel, pear, blueberry, horse, duck, hurricane, mosquito, snake, dolphin |
| Round Objects and Recreation | baseball, wheel, bush, crown, grass, lightning, snowman, church, soccer ball, rainbow, star, barn, basketball, leaf, moon, circle, rain, square, nose, compass, door, jail, tree, house, hospital, castle, smiley face, face, truck, book, the mona lisa, flower, snowflake, train, mountain, apple, cloud, nail, cannon, triangle, brain, clock, diamond, squiggle, river, line, beach |
| Electric Device | lighter, remote control, purse, keyboard, toaster, dresser, headphones, spreadsheet, washing machine, alarm clock, laptop, television, cooler, microwave, passport, envelope, teapot, power outlet, stove, paper clip, dishwasher, suitcase, cell phone, bucket, stereo, oven, calculator, mailbox |
| Common Objects | drill, frying pan, broom, spoon, baseball bat, hockey stick, scissors, rake golf club, matches, toothbrush, garden hose, paintbrush, crayon, bottlecap, sword, screwdriver, fork, knife, tennis racquet, paint can, hockey puck, syringe, shovel |
| Moving Objects | sailboat, airplane, skateboard, backpack, ambulance, sleeping bag, see saw, bench, flying saucer, aircraft carrier, clarinet, violin, canoe, ladder, cruise ship, school bus, bicycle, chair, submarine, speedboat, couch, motorbike, basket, drums, saxophone, trombone, tractor, police car, roller coaster, trumpet, swing set, firetruck, tent, cello, pickup truck, waterslide, boomerang |

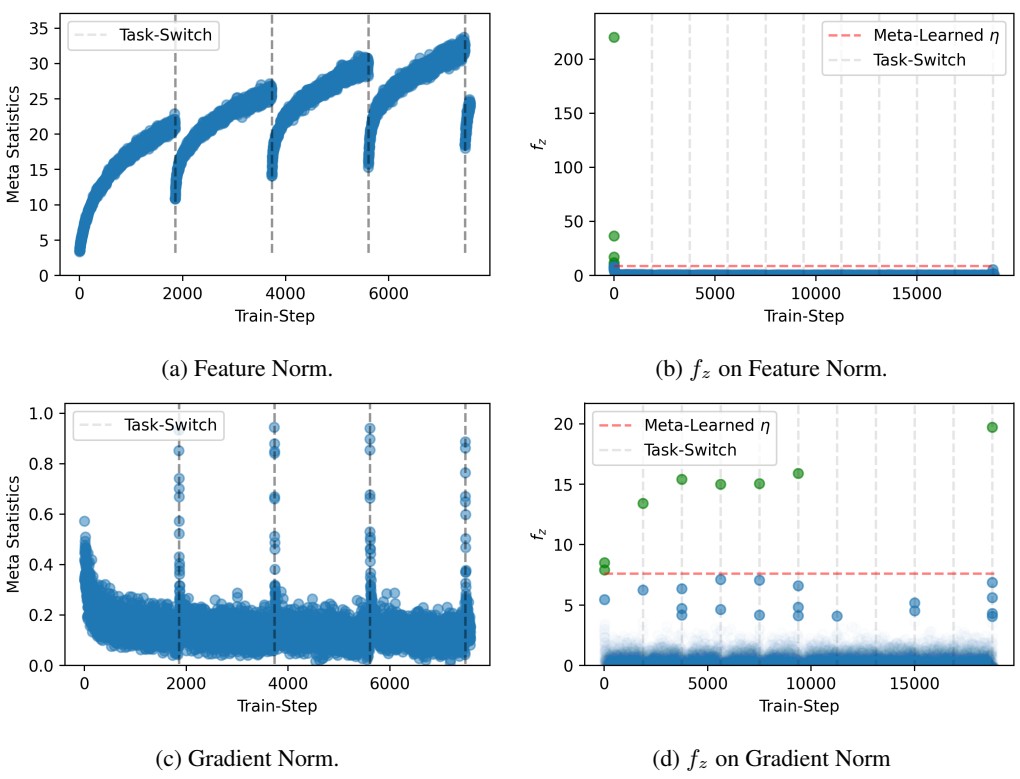

(a) Feature Norm.

(b) $f_z$ on Feature Norm.

(c) Gradient Norm.

(d) $f_z$ on Gradient Norm

Figure 6: We evaluate the informativeness of a Meta-Statistics such as the L2 norm of Features ($||\tau_F||_2$) (**top**) and the L2 norm of Gradients ($||\tau_G||_2$) (**bottom**) when used with a novelty detection method $f_z$ trained via our meta-learning process. On the **right** are the novelty scores for training batches where a score above $\eta$ (green markers) would be classified as a task transition. We find that an $f_z$ increases the linear separability of $||\tau_G||_2$ (*e.g.* comparison between sub-figures c $\rightarrow$ d) and as such makes it easy to discriminate a task switch where $||\tau_G||_2$ is more informative of a novel task transition. We find that not all MStats are informative of novelty, *e.g.* Figure 6b. A threshold can fail to generalize where the feature norm appears to be increasing Figure 6a, and is not linearly separable. Last, MStats such as gradient norm appear to generalize better between task sequences. We conclude that the choice of a MStats is important in the performance of a novelty detection method.

Table 15: Notation of Meta-Statistics and the processed MStats after dimensionality reduction.

| MStats ($\tau$) | Definition |
|---|---|
| Loss ($\tau_L$) (Aljundi et al., 2019a; Zhu et al., 2022) | $\mathcal{L}(f_\theta(x), y)$ |
| Gradient ($\tau_G$) (Huang et al., 2021) | $\nabla_\theta \mathcal{L}(f_\theta(x), y)$ |
| Fisher Info. ($\tau_{FI}$) (Park et al., 2022) | $\theta + [\nabla_\theta \mathcal{L}(f_\theta(x), \hat{y})]^2$ |
| Feature ($\tau_F$) (Tack et al., 2020) | $\phi(x)$ |
| Uncertainty ($\tau_U$) (Wortsman et al., 2020) | $\mathcal{H}(f_\theta(x))$ |
| Energy ($\tau_E$) (Liu et al., 2020) | $-\log \sum \exp(f_\theta(x))$ |
| **Processed MStats** | |
| $\overline{\tau_G}$ | Layer-wise gradient mean |
| $\|\tau_G\|_2$ | Layer-wise gradient L2 norm |
| $\overline{\tau_{Gx}}$ | Input gradient mean |
| $\overline{\tau_{G_{-1}}}$ | Pen-ultimate gradient mean |
| $\overline{\tau_{FI}}$ | Layer-wise Fisher Info. mean |
| $\|\tau_{FI}\|_2$ | Layer-wise Fisher Info. L2 norm |
| $\overline{\tau_F}$ | Layer-wise feature mean |
| $\|\tau_F\|_2$ | Layer-wise feature L2 norm |
| $\|\overline{\tau_F}\|_2$ | L2 norm of layer-wise feature mean |
| $\overline{\tau_{F_{-1}}}$ | Pen-ultimate feature mean |
| $\overline{\tau_U}$ | Uncertainty mean |
| $\overline{\tau_E}$ | Energy mean |

Table 16: (Part 1) F1 scores of novelty detection for 63 evaluation protocols. We exhaustively examine the possible combinations of MStats, for different processing and novelty detection methods (see Appendix C.1 and Table 15). We evaluated the aggregation functions using the mean[†] and max[‡] for a novelty detection method. We find that a fixed threshold may work well on standard benchmarks (PMNIST, SplitCIFAR) but cannot perform well on Stream benchmarks. We also observe that the F1 score is related to the design choices in the combination of settings we evaluate.

| Novelty Detection Learner | PMNIST | S-Num | S-Modal | SplitCIFAR | S-Num | S-Vis | Avg. F1 |
|---|---|---|---|---|---|---|---|
| | | Residual MLP | | | ResNet-18 | | |
| $f_z(\tau_L)$ | 1.000 | 0.848 | 0.974 | 0.947 | 0.973 | 0.947 | 0.948 |
| $f_z(\overline{\tau_G})^‡$ | 1.000 | 0.733 | 0.865 | 0.947 | 0.914 | 0.774 | 0.872 |
| $f_z(\|\tau_G\|_2)^†$ | 1.000 | 0.722 | 0.824 | 0.842 | 1.000 | 0.800 | 0.865 |
| $f_z(\overline{\tau_G})^†$ | 1.000 | 0.710 | 0.873 | 0.941 | 0.800 | 0.750 | 0.846 |
| $f_z(\overline{\tau_{FI}})^†$ | 1.000 | 0.516 | 0.473 | 0.947 | 0.973 | 0.593 | 0.750 |
| $f_z(\|\tau_G\|_2)^‡$ | 0.800 | 0.634 | 0.736 | 0.720 | 0.703 | 0.647 | 0.707 |
| $f_z(\overline{\tau_E})$ | 0.857 | 0.688 | 0.568 | 0.340 | 1.000 | 0.533 | 0.664 |
| $f_z(\overline{\tau_{FI}})^‡$ | 1.000 | 0.417 | 0.344 | 0.857 | 0.649 | 0.571 | 0.640 |
| $f_z(\|\tau_{FI}\|_2)^‡$ | 0.500 | 0.429 | 0.364 | 0.727 | 0.811 | 0.552 | 0.564 |
| $f_z(\overline{\tau_U})^†$ | 1.000 | 0.417 | 0.297 | 0.271 | 0.889 | 0.245 | 0.520 |
| $f_z(\|\overline{\tau_F}\|_2)^‡$ | 0.600 | 0.375 | 0.824 | 0.120 | 0.542 | 0.558 | 0.503 |
| $f_z(\|\tau_F\|_2)^‡$ | 0.600 | 0.400 | 0.847 | 0.069 | 0.511 | 0.462 | 0.481 |
| $f_z(\overline{\tau_{Gx}})^‡$ | 0.800 | 0.563 | 0.081 | 0.308 | 0.346 | 0.296 | 0.399 |
| $f_z(\overline{\tau_{G_{-1}}})^‡$ | 0.667 | 0.389 | 0.333 | 0.615 | 0.095 | 0.148 | 0.375 |
| $f_z(\overline{\tau_U})^‡$ | 0.444 | 0.303 | 0.286 | 0.203 | 0.698 | 0.304 | 0.373 |
| $f_z(\overline{\tau_F})^‡$ | 0.000 | 0.133 | 0.656 | 0.034 | 0.444 | 0.393 | 0.277 |
| $f_z(\overline{\tau_{G_{-1}}})^†$ | 0.250 | 0.036 | 0.111 | 0.706 | 0.233 | 0.308 | 0.274 |
| $f_z(\|\tau_{FI}\|_2)^†$ | 0.176 | 0.189 | 0.088 | 0.308 | 0.593 | 0.270 | 0.271 |
| $f_z(\overline{\tau_{Gx}})^†$ | 0.286 | 0.333 | 0.024 | 0.261 | 0.353 | 0.095 | 0.225 |
| $f_{sim}(\overline{\tau_F})$ | 0.429 | 0.261 | 0.329 | 0.005 | 0.023 | 0.000 | 0.174 |

Table 17: (Part 2) Table auxiliary to Table 16

| Novelty Detection Learner | PMNIST | S-Num | S-Modal | SplitCIFAR | S-Num | S-Vis | Avg. F1 |
|---|---|---|---|---|---|---|---|
| | | Residual MLP | | | ResNet-18 | | |
| $f_{\text{sim}}(\|\tau_F\|_2)$ | 0.400 | 0.103 | 0.177 | 0.090 | 0.125 | 0.095 | 0.165 |
| $f_{\text{sim}}(\|\overline{\tau_F}\|_2)$ | 0.400 | 0.108 | 0.166 | 0.108 | 0.138 | 0.068 | 0.165 |
| $f_{\text{sim}}(\|\tau_G\|_2)$ | 0.571 | 0.143 | 0.105 | 0.048 | 0.028 | 0.003 | 0.150 |
| $f_z(\overline{\tau_{F_{-1}}})^{\ddagger}$ | 0.029 | 0.051 | 0.418 | 0.017 | 0.042 | 0.030 | 0.098 |
| $f_z(\overline{\tau_F})^{\dagger}$ | 0.000 | 0.000 | 0.421 | 0.022 | 0.077 | 0.013 | 0.089 |
| $f_{\text{bayes}}(\overline{\tau_E})^{\dagger}$ | 0.012 | 0.024 | 0.016 | 0.089 | 0.038 | 0.040 | 0.036 |
| $f_z(\|\tau_F\|_2)^{\dagger}$ | 0.001 | 0.000 | 0.190 | 0.006 | 0.001 | 0.001 | 0.033 |
| $f_z(\|\overline{\tau_F}\|_2)^{\dagger}$ | 0.001 | 0.028 | 0.161 | 0.006 | 0.001 | 0.001 | 0.033 |
| $f_z(\overline{\tau_{F_{-1}}})^{\dagger}$ | 0.011 | 0.014 | 0.058 | 0.020 | 0.012 | 0.044 | 0.027 |
| $f_{\text{bayes}}(\|\tau_G\|_2)^{\dagger}$ | 0.007 | 0.022 | 0.016 | 0.065 | 0.016 | 0.013 | 0.023 |
| $f_{\text{bayes}}(\tau_L)^{\dagger}$ | 0.003 | 0.021 | 0.035 | 0.038 | 0.014 | 0.010 | 0.020 |
| $f_{\text{sim}}(\overline{\tau_{FI}})$ | 0.001 | 0.000 | 0.002 | 0.006 | 0.100 | 0.001 | 0.018 |
| $f_{\text{bayes}}(\|\overline{\tau_F}\|_2)^{\dagger}$ | 0.000 | 0.001 | 0.000 | 0.018 | 0.040 | 0.024 | 0.014 |
| $f_{\text{sim}}(\|\tau_{FI}\|_2)$ | 0.032 | 0.006 | 0.001 | 0.000 | 0.033 | 0.001 | 0.012 |
| $f_{\text{bayes}}(\overline{\tau_{FI}})^{\dagger}$ | 0.003 | 0.011 | 0.006 | 0.037 | 0.009 | 0.006 | 0.012 |
| $f_{\text{bayes}}(\overline{\tau_G})^{\dagger}$ | 0.005 | 0.011 | 0.006 | 0.037 | 0.006 | 0.005 | 0.012 |
| $f_{\text{bayes}}(\|\tau_{FI}\|_2)^{\dagger}$ | 0.002 | 0.009 | 0.002 | 0.031 | 0.006 | 0.007 | 0.010 |
| $f_{\text{sim}}(\overline{\tau_{F_{-1}}})$ | 0.008 | 0.000 | 0.025 | 0.012 | 0.006 | 0.000 | 0.008 |
| $f_{\text{bayes}}(\overline{\tau_U})^{\dagger}$ | 0.002 | 0.018 | 0.007 | 0.011 | 0.006 | 0.007 | 0.008 |
| $f_{\text{bayes}}(\overline{\tau_F})^{\dagger}$ | 0.007 | 0.001 | 0.003 | 0.007 | 0.017 | 0.014 | 0.008 |
| $f_{\text{bayes}}(\overline{\tau_{G_{-1}}})^{\dagger}$ | - | 0.013 | 0.003 | - | - | - | 0.008 |
| $f_{\text{bayes}}(\overline{\tau_{Gx}})^{\dagger}$ | 0.002 | 0.005 | 0.004 | 0.014 | 0.004 | 0.008 | 0.006 |
| $f_{\text{bayes}}(\overline{\tau_U})^{\ddagger}$ | 0.000 | 0.000 | 0.000 | 0.030 | 0.000 | 0.004 | 0.006 |
| $f_{\text{sim}}(\overline{\tau_U})$ | 0.001 | 0.003 | 0.001 | 0.023 | 0.001 | 0.000 | 0.005 |
| $f_{\text{sim}}(\overline{\tau_{G_{-1}}})$ | 0.002 | 0.006 | 0.005 | - | - | - | 0.004 |
| $f_{\text{sim}}(\overline{\tau_G})$ | 0.008 | 0.003 | 0.002 | 0.008 | 0.000 | 0.000 | 0.003 |
| $f_{\text{sim}}(\tau_{G_{-1}})$ | 0.002 | 0.001 | 0.001 | 0.012 | 0.002 | 0.001 | 0.003 |
| $f_{\text{bayes}}(\overline{\tau_F})^{\ddagger}$ | 0.000 | 0.001 | 0.001 | 0.013 | 0.002 | 0.000 | 0.003 |
| $f_{\text{bayes}}(\|\tau_F\|_2)^{\ddagger}$ | 0.002 | 0.002 | 0.003 | 0.006 | 0.003 | 0.001 | 0.003 |
| $f_{\text{sim}}(\overline{\tau_{Gx}})$ | 0.003 | 0.005 | 0.001 | 0.006 | 0.001 | 0.001 | 0.003 |
| $f_{\text{bayes}}(\overline{\tau_{F_{-1}}})^{\dagger}$ | 0.002 | 0.001 | 0.006 | 0.004 | 0.001 | 0.002 | 0.003 |
| $f_{\text{bayes}}(\|\overline{\tau_F}\|_2)^{\ddagger}$ | 0.002 | 0.003 | 0.003 | 0.004 | 0.001 | 0.001 | 0.002 |
| $f_{\text{bayes}}(\overline{\tau_{F_{-1}}})^{\ddagger}$ | 0.001 | 0.001 | 0.000 | 0.009 | 0.000 | 0.001 | 0.002 |
| $f_{\text{bayes}}(\tau_L)^{\ddagger}$ | 0.003 | 0.001 | 0.001 | 0.005 | 0.002 | 0.000 | 0.002 |
| $f_{\text{bayes}}(\overline{\tau_E})^{\ddagger}$ | 0.001 | 0.001 | 0.001 | 0.006 | 0.001 | 0.000 | 0.002 |
| $f_{\text{bayes}}(\overline{\tau_{Gx}})^{\ddagger}$ | 0.001 | 0.001 | 0.000 | 0.006 | 0.000 | 0.001 | 0.002 |
| $f_{\text{bayes}}(\overline{\tau_G})^{\ddagger}$ | 0.002 | 0.004 | 0.000 | 0.002 | 0.000 | 0.001 | 0.001 |
| $f_{\text{bayes}}(\overline{\tau_{FI}})^{\ddagger}$ | 0.004 | 0.000 | 0.000 | 0.003 | 0.000 | 0.000 | 0.001 |
| $f_{\text{bayes}}(\|\tau_G\|_2)^{\ddagger}$ | 0.002 | 0.000 | 0.000 | 0.004 | 0.000 | 0.000 | 0.001 |
| $f_{\text{bayes}}(\|\tau_F\|_2)^{\dagger}$ | 0.001 | 0.001 | 0.000 | 0.000 | 0.003 | 0.001 | 0.001 |
| $f_{\text{bayes}}(\|\tau_{FI}\|_2)^{\ddagger}$ | 0.002 | 0.002 | 0.000 | 0.000 | 0.000 | 0.001 | 0.001 |
| $f_{\text{sim}}(\tau_{F_{-1}})$ | 0.001 | 0.002 | 0.000 | 0.000 | 0.001 | 0.001 | 0.001 |
| $f_{\text{bayes}}(\overline{\tau_{G_{-1}}})^{\ddagger}$ | - | 0.001 | 0.000 | - | - | - | 0.001 |

Table 18: (Part 1) We use the Dispersion Ratio (VMR ↓) of the optimal threshold identified between different tasks for the same task sequence to evaluate the generalization of the identified threshold for the same 63 evaluation protocols from Table 16 and sort the rows by their F1 score. When comparing across task sequences, a lower VMR indicates an easier task sequence for a given Novelty Detection method. When comparing across Novelty Detection methods that perform similarly for the same dataset (in terms of F1 Score, *e.g.* the 2nd and 3rd rows in Table 16), a lower VMR indicates higher generalization for the method. We find that standard benchmarks ('PMNIST', 'SplitCIFAR') have lower VMRs, suggesting they are easier task sequences as the optimal threshold generalizes well among tasks within the same sequence. When comparing Novelty Detection methods that have discriminative power (listed in Table 18), they have higher VMR compared to less powerful methods. The result is an artifact of the high signal-to-noise ratio consequence of the informative MStats. While poor-performing Novelty Detection methods can produce extreme values for VMR consequence of an uninformative MStats. A VMR $> 1$ would indicate an over-dispersed distribution where a method is expected to generalize poorly. We conclude that Stream task sequences ('S-*') are more challenging to a threshold-based Novelty Detection method, while most methods are expected to generalize poorly (VMR $> 1$).

| Novelty Detection Learner | PMNIST | S-Num | S-Modal | SplitCIFAR | S-Num | S-Vis | Avg. VMR |
|---|---|---|---|---|---|---|---|
| | | Residual MLP | | | ResNet-18 | | |
| $f_z(\tau_L)$ | 0.137 | 1.080 | 0.597 | 0.187 | 0.842 | 0.721 | 1.444 |
| $f_z(\overline{\tau_G})^{\ddagger}$ | 0.144 | 1.050 | 0.677 | 0.192 | 0.679 | 0.583 | 0.866 |
| $f_z(\|\tau_G\|_2)^{\dagger}$ | 0.089 | 0.816 | 0.543 | 0.144 | 0.406 | 0.497 | 0.801 |
| $f_z(\overline{\tau_G})^{\dagger}$ | 0.168 | 1.147 | 0.621 | 0.304 | 0.742 | 0.767 | 1.124 |
| $f_z(\overline{\tau_{FI}})^{\dagger}$ | 0.165 | 1.207 | 0.819 | 0.248 | 1.078 | 0.870 | 1.343 |
| $f_z(\|\tau_G\|_2)^{\ddagger}$ | 0.145 | 0.892 | 0.556 | 0.179 | 0.527 | 0.512 | 0.784 |
| $f_z(\overline{\tau_E})$ | 0.160 | 0.835 | 0.687 | 0.260 | 0.545 | 0.477 | 0.915 |
| $f_z(\overline{\tau_{FI}})^{\ddagger}$ | 0.289 | 1.183 | 1.070 | 0.473 | 1.501 | 0.938 | 2.607 |
| $f_z(\|\tau_{FI}\|_2)^{\ddagger}$ | 0.343 | 1.331 | 1.418 | 0.301 | 1.717 | 0.939 | 3.542 |
| $f_z(\overline{\tau_U})^{\dagger}$ | 0.185 | 0.843 | 0.783 | 0.310 | 0.523 | 0.505 | 1.111 |
| $f_z(\|\overline{\tau_F}\|_2)^{\ddagger}$ | 0.079 | 0.892 | 2.358 | 0.288 | 0.560 | 0.341 | 2.147 |
| $f_z(\|\tau_F\|_2)^{\ddagger}$ | 0.082 | 0.939 | 2.626 | 0.285 | 0.651 | 0.314 | 2.564 |
| $f_z(\overline{\tau_{Gx}})^{\ddagger}$ | 0.246 | 0.600 | 0.509 | 0.242 | 0.663 | 0.592 | 1.698 |
| $f_z(\overline{\tau_{G_{-1}}})^{\ddagger}$ | 0.302 | 0.474 | 0.591 | 0.289 | 0.348 | 0.477 | 0.788 |
| $f_z(\overline{\tau_U})^{\ddagger}$ | 0.210 | 0.739 | 0.472 | 0.281 | 0.510 | 0.375 | 0.955 |

Table 19: (Part 2) Table auxiliary to Table 18

| Novelty Detection Learner | PMNIST | S-Num Residual MLP | S-Modal | SplitCIFAR | S-Num ResNet-18 | S-Vis | Avg. VMR |
|---|---|---|---|---|---|---|---|
| $f_z(\overline{\tau_F})^\ddagger$ | 0.430 | 1.659 | 1.155 | 0.244 | 0.663 | 0.432 | 1.376 |
| $f_z(\overline{\tau_{G_{-1}}})^\dagger$ | 13.535 | 24.815 | 28.018 | 22.213 | 25.257 | 41.921 | 45.353 |
| $f_z(||\tau_F I||_2)^\dagger$ | 0.194 | 0.934 | 0.616 | 0.146 | 1.239 | 0.676 | 2.073 |
| $f_z(\overline{\tau_{Gx}})^\dagger$ | 7.674 | 153.825 | 19.870 | 3.762 | 18.381 | 50.809 | 420.020 |
| $f_{\mathrm{sim}}(\overline{\tau_F})$ | 0.546 | 1.384 | 0.835 | 1.233 | 1.502 | 1.380 | 1.525 |
| $f_{\mathrm{sim}}(||\tau_F||_2)$ | 0.623 | 1.934 | 1.651 | 3.651 | 1.456 | 1.008 | 2.479 |
| $f_{\mathrm{sim}}(||\overline{\tau_F}||_2)$ | 0.518 | 1.844 | 1.597 | 0.606 | 0.693 | 0.629 | 2.445 |
| $f_{\mathrm{sim}}(||\tau_G||_2)$ | 0.314 | 0.998 | 1.003 | 0.967 | 0.598 | 0.640 | 1.056 |
| $f_z(\overline{\tau_{F_{-1}}})^\ddagger$ | 0.208 | 0.440 | 0.523 | 0.149 | 0.347 | 0.297 | 0.528 |
| $f_z(\overline{\tau_F})^\dagger$ | 1.557 | 3.251 | 969.710 | 1.606 | 4.317 | 1.791 | 12.803 |
| $f_{\mathrm{bayes}}(\overline{\tau_E})^\dagger$ | 0.146 | 0.152 | 0.206 | 0.197 | 0.272 | 0.230 | 0.358 |
| $f_z(||\tau_F||_2)^\dagger$ | 0.077 | 1.392 | 17.899 | 0.668 | 0.746 | 0.455 | 5.179 |
| $f_z(||\overline{\tau_F}||_2)^\dagger$ | 0.071 | 1.109 | 2870.817 | 0.604 | 0.699 | 0.444 | 3.302 |
| $f_z(\overline{\tau_{F_{-1}}})^\dagger$ | 2.682 | 5.162 | 8.375 | 1.115 | 18.252 | 1.002 | 4.760 |
| $f_{\mathrm{bayes}}(||\tau_G||_2)^\dagger$ | 0.031 | 0.032 | 0.035 | 0.050 | 0.060 | 0.076 | 0.249 |
| $f_{\mathrm{bayes}}(\tau_L)^\dagger$ | 0.104 | 0.216 | 0.123 | 0.120 | 0.293 | 0.128 | 0.269 |
| $f_{\mathrm{sim}}(\overline{\tau_{FI}})$ | - | 13.416 | 7.211 | - | 3.364 | - | 6.814 |
| $f_{\mathrm{bayes}}(||\overline{\tau_F}||_2)^\dagger$ | 0.085 | 0.546 | 1.132 | 0.099 | 0.203 | 0.125 | 2.116 |
| $f_{\mathrm{sim}}(||\tau_{FI}||_2)$ | 8.718 | 5.181 | 3.004 | 9.219 | 6.300 | - | 4.504 |
| $f_{\mathrm{bayes}}(\overline{\tau_{FI}})^\dagger$ | 0.014 | 0.028 | 0.015 | 0.010 | 0.053 | 0.015 | 0.459 |
| $f_{\mathrm{bayes}}(\overline{\tau_G})^\dagger$ | 0.009 | 0.031 | 0.010 | 0.010 | 0.064 | 0.011 | 0.445 |
| $f_{\mathrm{bayes}}(||\tau_F I||_2)^\dagger$ | 0.019 | 0.031 | 0.036 | 0.011 | 0.088 | 0.018 | 0.403 |
| $f_{\mathrm{sim}}(\overline{\tau_{F_{-1}}})$ | 0.433 | 1.169 | 0.916 | 0.231 | 0.724 | 0.853 | 1.422 |
| $f_{\mathrm{bayes}}(\overline{\tau_U})^\dagger$ | 0.021 | 0.079 | 0.176 | 0.025 | 0.123 | 0.041 | 0.156 |
| $f_{\mathrm{bayes}}(\overline{\tau_F})^\dagger$ | 0.042 | 0.089 | 0.179 | 0.033 | 0.087 | 0.039 | 0.588 |
| $f_{\mathrm{bayes}}(\overline{\tau_{G_{-1}}})^\dagger$ | - | 0.020 | 0.004 | - | - | - | 0.018 |
| $f_{\mathrm{bayes}}(\overline{\tau_{Gx}})^\dagger$ | 0.016 | 0.049 | 0.031 | 0.032 | 0.100 | 0.022 | 0.593 |
| $f_{\mathrm{bayes}}(\overline{\tau_U})^\ddagger$ | 0.021 | 0.079 | 0.176 | 0.025 | 0.123 | 0.041 | 0.156 |
| $f_{\mathrm{sim}}(\overline{\tau_U})$ | 0.224 | 0.896 | 1.064 | 0.272 | 0.707 | 0.682 | 1.547 |
| $f_{\mathrm{sim}}(\overline{\tau_{G_{-1}}})$ | 0.033 | 0.031 | 0.033 | - | - | - | 0.033 |
| $f_{\mathrm{sim}}(\overline{\tau_G})$ | 0.084 | 0.110 | 0.131 | 0.062 | 0.089 | 0.067 | 0.110 |
| $f_{\mathrm{sim}}(\tau_{G_{-1}})$ | 0.013 | 0.080 | 0.027 | 0.005 | 0.037 | 0.020 | 0.042 |
| $f_{\mathrm{sim}}(\tau_{G_{-1}})$ | 0.013 | 0.080 | 0.027 | 0.005 | 0.037 | 0.020 | 0.042 |
| $f_{\mathrm{bayes}}(\overline{\tau_F})^\ddagger$ | 0.042 | 0.089 | 0.179 | 0.033 | 0.087 | 0.039 | 0.588 |
| $f_{\mathrm{bayes}}(||\tau_F||_2)^\ddagger$ | 0.235 | 0.657 | 1.149 | 0.286 | 0.824 | 0.507 | 2.121 |
| $f_{\mathrm{sim}}(\overline{\tau_{Gx}})$ | 0.032 | 0.038 | 0.035 | 0.009 | 0.035 | 0.020 | 0.032 |
| $f_{\mathrm{bayes}}(\overline{\tau_{F_{-1}}})^\dagger$ | 0.043 | 0.127 | 0.085 | 0.114 | 0.172 | 0.218 | 1.025 |
| $f_{\mathrm{bayes}}(||\overline{\tau_F}||_2)^\ddagger$ | 0.085 | 0.546 | 1.132 | 0.099 | 0.203 | 0.125 | 2.116 |
| $f_{\mathrm{bayes}}(\overline{\tau_{F_{-1}}})^\ddagger$ | 0.043 | 0.127 | 0.085 | 0.114 | 0.172 | 0.218 | 1.025 |
| $f_{\mathrm{bayes}}(\tau_L)^\ddagger$ | 0.104 | 0.216 | 0.123 | 0.120 | 0.293 | 0.128 | 0.269 |
| $f_{\mathrm{bayes}}(\overline{\tau_E})^\ddagger$ | 0.146 | 0.152 | 0.206 | 0.197 | 0.272 | 0.230 | 0.358 |
| $f_{\mathrm{bayes}}(\overline{\tau_{Gx}})^\ddagger$ | 0.016 | 0.049 | 0.031 | 0.032 | 0.100 | 0.022 | 0.593 |
| $f_{\mathrm{bayes}}(\overline{\tau_G})^\ddagger$ | 0.009 | 0.031 | 0.010 | 0.010 | 0.064 | 0.011 | 0.445 |
| $f_{\mathrm{bayes}}(\overline{\tau_{FI}})^\ddagger$ | 0.014 | 0.028 | 0.015 | 0.010 | 0.053 | 0.015 | 0.459 |
| $f_{\mathrm{bayes}}(||\tau_G||_2)^\ddagger$ | 0.031 | 0.032 | 0.035 | 0.050 | 0.060 | 0.076 | 0.249 |
| $f_{\mathrm{bayes}}(||\tau_F||_2)^\dagger$ | 0.235 | 0.657 | 1.149 | 0.286 | 0.824 | 0.507 | 2.121 |
| $f_{\mathrm{bayes}}(||\tau_{FI}||_2)^\ddagger$ | 0.019 | 0.031 | 0.036 | 0.011 | 0.088 | 0.018 | 0.403 |
| $f_{\mathrm{sim}}(\tau_{F_{-1}})$ | 0.044 | 0.133 | 0.055 | 0.048 | 0.057 | 0.096 | 0.119 |
| $f_{\mathrm{bayes}}(\overline{\tau_{G_{-1}}})^\ddagger$ | - | 0.020 | 0.004 | - | - | - | 0.018 |

