# OpenReview forum: "Stream: A Generalized Continual Learning Benchmark and Baseline"
_ICLR.cc/2024/Conference — ICLR 2024 Conference Withdrawn Submission_

### Official Review · Reviewer_AAR3 · 2023-10-20

**Soundness:** 3 good
**Presentation:** 2 fair
**Contribution:** 2 fair
**Rating:** 5
**Confidence:** 4

**Summary:**

This paper presents a multi-modal benchmark for general continual learning (GCL), i.e., task identity not used in training. The proposed benchmark constructs a sequence of tasks with varying learning-gaps from vision and text datasets, through mapping them into the same feature space and augmenting them within the feature space. The authors then propose a method based on meta-learning to perform novelty detection in GCL, which achieves some performance improvements.

**Strengths:**

1. The proposed benchmark is interesting. It combines the challenges of multi-modality and GCL, and enables an in-depth analysis of learning-gaps in continual learning.
2. The authors present good analyses, such as the t-sne of multi-modal distributions in Figure 2 and the summary of novelty detection scores in Table 2.
3. The proposed method seems to be reasonable, and achieves some performance improvements on ER.

**Weaknesses:**

1. I think the multi-modal stream (i.e., S-Modal) is the primary contribution of this paper. The single-modal streams (i.e., S-Num and S-Vis) seem to be less informative, since the considered datasets (MNIST, SVHN, CIFAR-10, CINIC-10) are relatively simple and widely-used in continual learning.
2. The most primary concern is that, although I appreciate the idea of multi-modal stream, it requires large pre-trained backbones for feature mapping and artificially synthesizes the projected features. Since the variation of learning-gaps usually happen in data space, although they are then projected to feature space, such "natural variation" and the proposed "artificial variation" may not be consistent. In other words, the proposed benchmark may not reflect the realistic dynamics. Besides, the pre-trained models are usually adapted through prompting or other adaptation techniques, which potentially enlarge this issue.
3. The proposed method requires meta-learning on a dummy stream, which potentially assume that the data distributions of meta-learning and GCL are similar. Have the authors considered the dissimilar scenario?
4. The authors claim that they beat current state-of-the-art CL approaches, which might be too strong. They have only compared with three classic baselines and achieve moderate improvements.

**Questions:**

Please refer to Weakness.

---

### Official Review · Reviewer_18Hf · 2023-10-30

**Soundness:** 2 fair
**Presentation:** 2 fair
**Contribution:** 2 fair
**Rating:** 3
**Confidence:** 3

**Summary:**

This paper introduces a General Continual Learning (GCL) benchmark called Stream. It considers non-uniform learning-gaps across vision and text datasets. This paper also introduces a baseline on the proposed benchmark, named αMetaSup. Extensive analysis are shown in this paper.

**Strengths:**

1. The Stream benchmark introduced in this work is meaningful for GCL. It provides this task with more realistic, large-scale datasets with infinitely long task sequences and non-uniform learning gaps.
2. Considering meta-learning to detect novelty for GCL is a reasonable baseline.
3. The proposed method, αMetaSup, shows a significant improvement in the Multi-Modal setting (S-Modal).
4. This work evaluates a wide range of design choices.
5. Transforming datasets within the same feature space sounds interesting.

**Weaknesses:**

1. The organization of this paper needs to be refined. This paper tries to include as many "contributions" (Benchmark, transformation space, transformation method, new novel detection method, new sampling method, etc.) as possible, making it very hard to tell whether all contributions are of high quality. I am not sure whether all claims are well supported by experiments and details since there are too many claims in this paper, but the ablations are not well organized to support each claim.

2. About the method, it said that αMetaSup trains a dummy learner on the Digital classification setting (S-Num) and then uses it on S-Modal. Why would this meta-learned learner be suitable for the complex S-Modal setting? Just as indicated in Table 1, S-Num (MNIST and SVHN) and S-Vis (CIFAR-10 and CINIC-10) settings are much easier than S-Modal (DomainNet, IMDb and Amazon). I do not think the conclusion from this meta-transfer result is universal for GCL.

3. In the experiments, it is reported that "Meta-Sampling improves performance by as much as 10.5% and 9.6% on S-Num and S-Vis.", "αMetaSup augmented baseline ER performs 8% better."  How do these numbers be obtained? I think they are not matched with the results reported in the Table.

4. Presentation of this work also needs to be refined.
For instance,
- Stream is proposed as a benchmark but is also stated as "we propose Stream as a method", which is ambiguous.
- "We perform a systematic analysis of meta-training statistics from the literature that are used to identify novel tasks, to find that they correlate to the learning-gap." The meta-training statistics are known for identifying novel tasks in the past literature. Why (the analysis is) to find that they correlate to the learning-gap"? Similar sentences can be found, such as "We survey Meta-Statistics (Table 2) from the literature to show the novelty of our benchmark and when compared with previous benchmarks." Why can the survey show the novelty of the benchmark?
- In Table 5, the methods in the last two rows are supposed to be "f_{h} + Meta-Samp + ER" and "αMetaSup + Meta-Samp + ER", right?

**Questions:**

See Weaknesses part for the questions.
Will the codes for constructing Stream benchmark be open-sourced?

---

### Official Review · Reviewer_uFkC · 2023-10-31

**Soundness:** 2 fair
**Presentation:** 3 good
**Contribution:** 2 fair
**Rating:** 3
**Confidence:** 3

**Summary:**

The paper proposes Stream that is method which continuosly increases the sequence of tasks. The proposed method creates tasks such that the distribution between consecutive tasks change. The authors perform a review of commonly used Meta-Statistics that are used to identify new tasks over time. In addition, the authors propose αMetaSup that is a method which identifies novel task transitions.

**Strengths:**

The proposed benchmark shows weaknesses of current general continual learning methods.

The authors identify knowledge gaps of existing methods.

The authors use multiple statistical metrics and perform extensive numerical results.

**Weaknesses:**

My major concern with this paper is whether a realistic scenario is being put forward.

The contributions of the paper are not clear and the significance of the proposed benchmark is not clear.

**Questions:**

Is Stream a realistic scenario?

Do the authors randomly alter the order of the tasks in the numerical results? As you say in the paper, methods can perform differently when trained on the same set of tasks in a different order.

Can you include the metric backward transfer (BWT) [1] to evaluate the forgetting?

[1] Gradient Episodic Memory for Continual Learning

---

### Official Review · Reviewer_Tzaa · 2023-11-01

**Soundness:** 3 good
**Presentation:** 3 good
**Contribution:** 2 fair
**Rating:** 5
**Confidence:** 4

**Summary:**

A synthetic multi-modal benchmark for Generalised Continual Learning (GCL), Stream, and a baselines method using a Transformer to identify novel task transitions, αMetaSup.

**Strengths:**

- The field of continual learning would benefit from contributions in benchmarks and solid baselines in which this work aims to address.

**Weaknesses:**

- Stream is yet another synthetic benchmark that uses transformations to create a "large" sequence of "novel" tasks. It has not been shown that these transformations yield realistically novel tasks, where the gap can be trivially learned. In other words, the realism of this novelty has not been demonstrated.
- It is not clear why existing synthetic benchmarks can't be used ignoring the task label.
- No empirical comparison to existing synthetic benchmarks was conducted (see previous point).
- The proposed baseline was not validated using other benchmarks.
- It is not clear if the methods developed on this benchmark would generalise to real-world data.
- The datasets involved in Stream are at this point trivial for a new benchmark.

**Questions:**

See Weaknesses Section above.